# Rapid and robust restoration of breathing long after spinal cord injury

Philippa M. Warren [ID] [1,2,3], Stephanie C. Steiger[4], Thomas E. Dick[1,5], Peter M. MacFarlane[6], Warren J. Alilain[1,7] & Jerry Silver[1]

There exists an abundance of barriers that hinder functional recovery following spinal cord injury, especially at chronic stages. Here, we examine the rescue of breathing up to 1.5 years following cervical hemisection in the rat. In spite of complete hemidiaphragm paralysis, a single injection of chondroitinase ABC in the phrenic motor pool restored robust and persistent diaphragm function while improving neuromuscular junction anatomy. This treatment strategy was more effective when applied chronically than when assessed acutely after injury. The addition of intermittent hypoxia conditioning further strengthened the ventilatory response. However, in a sub-population of animals, this combination treatment caused excess serotonergic (5HT) axon sprouting leading to aberrant tonic activity in the diaphragm that could be mitigated via 5HT2 receptor blockade. Through unmasking of the continuing neuroplasticity that develops after injury, our treatment strategy ensured rapid and robust patterned respiratory recovery after a near lifetime of paralysis.

[1] Department of Neurosciences, Case Western Reserve University, 10900 Euclid Avenue, Cleveland, OH 44106, USA. [2] School of Biomedical Sciences, Faculty of Biological Sciences, University of Leeds, Leeds LS2 9JT, UK. [3] The Wolfson Centre for Age Related Diseases, Institute of Psychiatry, Psychology and Neuroscience (IoPPN), King's College London, London SE1 1UL, UK. [4] Department of Biology, Case Western Reserve University, 10900 Euclid Avenue, Cleveland, OH 44106, USA. [5] Division of Pulmonary Critical Care and Sleep Medicine, Department of Medicine, Case Western Reserve University, 10900 Euclid Avenue, Cleveland, OH 44106, USA. [6] Department of Paediatrics, Rainbow Babies & Children's Hospital, Case Western Reserve University, 10900 Euclid Avenue, Cleveland, OH 44106, USA. [7] Spinal Cord and Brain Injury Research Centre, Department of Neuroscience, University of Kentucky, Lexington, KY 40536, USA. Correspondence and requests for materials should be addressed to P.M.W. (email: pmw45@case.edu) or to W.J.A. (email: warren.alilain@uky.edu) or to J.S. (email: jxs10@case.edu)

The ability to restore function following lengthy, paralyzing spinal cord injury (SCI) is a daunting prospect. The success of acutely applied treatments is rarely replicated chronically[1,2] in part due to axonal entrapment within the glial scar, reduced levels of regeneration-associated genes and proteins[3], and maintenance of a plasticity dampening perineuronal net (PNN)[4]. Modest anatomical regeneration of motor and sensory axons is possible a year following injury[5], particularly when intrinsic neuronal growth is stimulated via *Pten* deletion[6] or when neurotrophic factors are combined with peripheral nerve grafting[7]. However, restoration of essential motor systems at extended time points after SCI has long proved elusive. This has led to the long-standing belief that robust functional recovery chronically is far more difficult, if not impossible, to achieve.

Over 50% of human SCIs cause deficits in respiratory motor and ventilatory function[8,9]. The inability to breathe is the major cause of morbidity and mortality in human SCI patients[9]. Marginal (~10% of pre-injury levels) restoration of respiratory muscle activity following acute cervical hemisection has been achieved utilizing chondroitinase ABC (ChABC)[10]. This enzyme catabolizes the chondroitin sulphate proteoglycans (CSPGs) of the protective, yet growth inhibitory, scar and PNNs, facilitating axonal regeneration or sprouting and enhancing synaptic strength[11,12]. These functional gains are partially caused by increased sprouting of 5HT fibres[13]. Combining ChABC with task-specific rehabilitation after SCI is known to increase locomotor ability by augmenting and structuring plasticity[14,15]. Similar to locomotor training, acute repetitive intermittent hypoxia (IH) conditioning can facilitate a small, transient gain in respiratory function following sub-chronic SCI[16]. We speculated whether a combined ChABC and IH treatment might achieve a meaningful degree of long-lasting respiratory motor repair at protracted stages after cervical hemisection. At extended time points post-injury, our strategy produced rapid and robust respiratory motor recovery which was largely dependent on matrix degradation. The recovered activity was persistent over time and greatly superior to that which occurred acutely.

## Results

**No spontaneous hemi-diaphragm recovery.** Adult rats received a lateral cervical (C) level C2 hemisection. This injury severs ipsilateral glutamatergic and serotonergic projections from the brainstem (Supplementary Figure 1a)[17], abolishing the major descending inputs to the phrenic motor pool (PMP). This extensive injury damages areas of the spinal cord containing pre-phrenic interneuron pathways and direct projections which ultimately innervate the PMP (Supplementary Figure 1d-i). At its epicentre, the slightly over-hemisected lesion encompassed 51.2% of the total spinal cord volume (Supplementary Figure 1h). Quantitative analysis of lesion volumetrics from a set region around the injury showed minimal divergence in the lesion area between animals (analysis of variance; ANOVA, $p = 0.0781$; $F(5,15) = 2.49$). There was no evidence that regeneration occurred through the site of injury. 49.8% of animals demonstrated minimal ipsilateral hemidiaphragm activity during nasal occlusion at baseline (Supplementary Figure 1j-l). This form of ventilatory stimulation strongly increases respiratory drive, but not through pathways associated with eupneic breathing[18,19] (Supplementary Figure 1j-k). Importantly, having a minimal response to transient nasal occlusion did not predict which animals would recover following treatment (Supplementary Figure 1l). In our hands, this anatomically and physiologically complete injury results in total and persistent, ipsilateral hemidiaphragm paralysis during eupnea at all chronic time points (Figs. 1d–g, 2b–g, 3b–e and 6b, c, e, f; see below).

**ECM breakdown restores respiratory function.** Treatment was started 12 weeks after the hemisection (Supplementary Table 1, group 1). Following baseline diaphragm electromyography (dia-EMG) recordings, animals received a single injection of either ChABC or vehicle into the ipsilateral C4 PMP (Fig. 1a). As such, we were not trying to aid regeneration through the site of injury but, rather, facilitate sprouting from the contralateral side in order to augment the recovery of function at the cervical level where it is likely to be required. Conditioning with IH or air (normoxia; Nx) was delayed for 7 days. Mimicking an established daily protocol[16] (Fig. 1a), conditioning/air breathing occurred for 3 weeks before final diaEMG recordings.

Treatment of the chronic injury with vehicle and air/IH caused no functional restoration of the ipsilateral hemidiaphragm or compensatory increases in contralateral diaEMG amplitude (Fig. 1b–e; Supplementary Movies 1 and 2). Plasticity induced by IH conditioning has previously been shown to aid locomotor function (see below)[16,20]. However, we demonstrate that, by itself, it did not manifest significant effects upon eupnoeic pulmonary activity in long-term, completely hemiplegic subjects. IH conditioning caused significant increases in glial fibrillary acidic protein (GFAP) immunoreactivity within the ipsilateral (ANOVA, $p = 0.0003$, $F = 13.73$) and contralateral (ANOVA, $p = 0.0065$, $F = 6.74$) PMPs (Supplementary Figure 2b-f and aa-ae), confirming that conditioning can stimulate astrocytic reactivity[21]. Wisteria floribunda agglutinin (WFA) staining revealed that the IH-induced increase in GFAP did not correlate with an augmentation in PNNs (Fig. 1w–aa; Supplementary Figure 2v-z).

Application of ChABC (with or without IH) resulted in the restoration of synchronized diaphragm function ipsilateral to the injury in 66.7% of the animals (Fig. 1b, f, g; ANOVA, $p < 0.0001$, $F = 38.79$). The restored activity was far greater than that achieved via the same treatment applied acutely post SCI[10] and was not statistically different from that of the contralateral hemidiaphragm (Supplementary Movie 3). These data show that CSPG removal is both necessary and sufficient to restore respiratory motor function at chronic stages after SCI. This result has been replicated three times during various experiments (Supplementary Table 1), demonstrating the accuracy and validity of the data in a large number of animals. Following ChABC treatment in our first cohort, the contralateral hemidiaphragm functioned at ~60% of that generated during airway occlusion (Fig. 1c). This percentage corresponds to values typical of an uninjured and anaesthetised animal[22]. As our injured animals are unlikely to be hypercapnic or hypoxic[23], these data suggest that chronic motor system behavioural compensation acts to ensure respiratory activity responds at physiologic levels during eupnea, potentially through an increase in respiration rate[23]. Following ChABC-mediated ECM breakdown with air, the ipsilateral hemidiaphragm functioned at ~30% of that during airway occlusion (Fig. 1b). When ChABC was combined with IH, diaphragm activity increased to ~40%. As such, IH conditioning conferred modest benefit to the ChABC-mediated treatment strategy, possibly due to increased neuroplasticity[16]. Indeed, the combination treatment promoted coordinated motor activity indistinguishable from normal respiration (Supplementary Movie 4). Importantly, restoration of function occurred rapidly following the induction of treatment. Two of the combination treated animals showed unique tonic firing of the ipsilateral hemidiaphragm (star data points, Fig. 1b; see Fig. 4 and further discussion below). The frequency of breaths was not statistically different between treatment conditions (Supplementary Figure 2a; ANOVA, $p = 0.1422$, $F = 1.821$) indicating that respiratory control mechanisms emanating from the lower brainstem remain intact following the induction of plasticity in the chronically injured animal[24].

33.3% of the animals in both the ChABC treatment groups did not respond to the enzyme (triangle data points, Fig. 1b). Enzyme activity was confirmed through WFA (ANOVA, ipsilateral $p = 0.013$, $F = 5.54$; contralateral $p = 0.0049$, $F = 7.27$) and 2B6 staining (ANOVA, ipsilateral $p = 0.015$, $F = 5.33$; contralateral $p = 0.018$, $F = 4.97$) at the level of the PMPs (Fig. 1r–aa; Supplementary Figure 2q-z) demonstrating chondroitin sulphate removal from both the ECM and PNN. Further, immunohistochemistry showed that TrkB increased in response to IH treatment (Fig. 1m–q; Supplementary Figure 2l-p) in the ipsilateral (ANOVA, $p = 0.0411$, $F = 3.76$) and contralateral (ANOVA, $p = 0.0015$; $F = 9.78$) PMPs. These data confirm that IH conditioning had manifested its well-known effects upon receptor expression at the PMP[25]. 5HT also increased following IH treatment on the ipsilateral side of the cord (Fig. 1h–l; ANOVA, $p = 0.0411$, $F = 3.76$). However, these 5-HT and receptor increases did not result in functional recovery. Nonetheless, the amount of 5HT was positively correlated with the activity amplitude of the ipsilateral hemidiaphragm (Supplementary Table 2; coefficient of determination, $R^2 = 0.228$, $p = 0.045$), suggesting that enzyme treatment in non-responding animals did not induce enough 5HT sprouting to facilitate positive gains in function. In support of this hypothesis we found that removal of CSPGs was strongly correlated with increasing ipsilateral hemidiaphragm amplitude (Supplementary Table 2; coefficient of determination, $R^2 = 0.698$, $p = 0.00005$) and that there was a positive correlation between 5HT staining intensity and that of both 2B6 (coefficient of determination, $R^2 = 0.524$, $p = 0.002$) and WFA (coefficient of determination, $R^2 = 0.302$, $p = 0.027$) at the ipsilateral PMP (Supplementary Table 2). Interestingly, upregulation of TrkB also showed a positive correlation with increased amplitude on the ipsilateral side of the cord (Supplementary Table 2; coefficient of determination, $R^2 = 0.477$, $p = 0.003$). However, TrkB staining intensity did not correlate well with CSPG removal (2B6: coefficient of determination, $R^2 = 0.125$, $p = 0.178$; WFA: $R^2 = 0.077$, $p = 0.298$), suggesting that increases in this receptor may have been facilitated by IH-induced plasticity. As only animals with CSPG removal show recovery of function, increases in 5HT may be deemed more significant to recovery than TrkB. No other correlations on either the ipsi- or contralateral side of the cord yielded a significant result (Supplementary Table 2). These data suggest that robust restoration of respiratory function following chronic SCI could be dependent, but only in part, upon levels of endogenous serotonin (Fig. 1h–l).

**Persistent breathing restoration by CSPG breakdown.** To examine the magnitude of the restorative effect that ECM degradation and optimal respiratory conditioning had upon respiratory motor behaviour, 12 weeks after C2 hemisection, additional groups of animals were all injected with ChABC (Fig. 2a; Supplementary Table 1, group 2). The activity of the enzyme at the PMP was confirmed through 2B6 and WFA staining (Supplementary Figure 3b-c,f-g,h-i). IH conditioning or air were then administered for 1, 3, or 5 weeks.

The degree of functional restoration achieved in all groups was significant (ANOVA, $p < 0.0001$; $F = 14.43$). Again, 66.7% of the animals in each group demonstrated strong, synchronized activity in the ipsilateral hemidiaphragm (Fig. 2b–h). Recovery was maximal in as little as 2 weeks following administration of the enzyme and as brief as 1 week exposure to IH (Fig. 2b, c). The amplitude of activity in all groups was statistically similar to that exhibited in the contralateral hemidiaphragm (Fig. 2h, i). Animals treated with ChABC and air achieved ipsilateral diaphragm activity which functioned at ~30% of that recruited during occlusion. This

activity occurred at ~57% in all animals treated with ChABC and IH, analogous to the uninjured animal (Fig. 2h, i)[22].

The rapid functional improvement at all time points likely reflects sprouting or, more likely, the sustained unmasking of ongoing neuroplasticity engendered around the ipsilateral PMP and, particularly, 5HT-positive fibres (Fig. 2j–p; ANOVA, $p = 0.404$, $F = 1.08$). Indeed, 66.66% of animals treated with either ChABC or ChABC+IH for 6 weeks (groups 2e+f and 4c+d) which had not responded to treatment demonstrated activity within the previously paralysed hemidiaphragm upon bolus application of 5HT (Fig. 2q–s). These data highlight that removing the inhibitory matrix around the PMP, resulting in increased levels of 5HT, is necessary and sufficient to enable restoration of respiratory motor function following chronic injury. Increases in TrkB receptors were also sustained (Supplementary Figure 3a,e; ANOVA; ipsilateral: $p = 0.235$, $F = 1.51$; contralateral: $p = 0.313$, $F = 1.28$) aiding recovery at all acute time points following treatment. During this time, the PNN was gradually returning within the ipsilateral phrenic nucleus (Supplementary Figure 3c) signifying how these treatment-induced anatomical alterations were slowly being incorporated into the spinal cord environment. The frequency of breaths was, again, not statistically different between treatment conditions (Supplementary Figure 3j; ANOVA, $p = 0.2247$, $F = 1.406$).

To assess the effect our treatment strategy had upon the respiratory motor system as a whole, we assessed diaphragm neuromuscular junctions (diaNMJs) (Supplementary Figure 4a-i). Animals treated with ChABC and 5 weeks of IH conditioning or air, showed higher numbers of intact diaNMJs across the ipsilateral hemidiaphragm (ANOVA, ventral $p = 0.0093$, $F = 7.16$; medial $p = 0.0356$, $F = 4.44$; dorsal $p = 0.0121$, $F = 6.56$; Supplementary Table 1, group 4). This correlated with clear decreases in complete (ANOVA, ventral $p = 0.0049$, $F = 8.74$) and partially (ANOVA, ventral $p = 0.0117$, $F = 6.64$; medial $p = 0.0106$, $F = 6.86$; dorsal $p = 0.0106$, $F = 6.86$) denervated junctions in these groups. We also assessed the effect that ECM degradation had upon the functional capacity of the diaphragm at this time point in the same animals. Plethysmography data revealed that IH (in either saline and ChABC injected animals) did not significantly enhance ventilatory responses ($V_E$) at early (6 week) time points post-treatment. We did, however, observe improvements following ChABC treatment in both the hypoxic and hypercapnic ventilatory responses (HVR and HCVR) and in both cases the effects were primarily through alterations in frequency (Supplementary Figure 5a–c). Especially notable, however, was the more robust ventilatory capacity (specifically the HVR, Supplementary Figure 5b) in the longer-term animals (3–6 months post-treatment; Supplementary Table 1, group 3, see below) which was mediated through changes in frequency and tidal volume (VT). At these lengthy time points, treatment with ChABC alone, or in combination with IH, revealed beneficial characteristics (Supplementary Figure 5d–f). These data are important as they were conducted in freely moving unanaesthetised animals. As all responses to baseline conditions (air) were statistically non-divergent between groups (Supplementary Table 3), these data suggest that ChABC treatment can facilitate an extended and beneficial return of function in injured animals.

We had established that CSPG breakdown and conditioning mediates restoration of respiratory function following chronic diaphragm paralysis. However, the potential permanence of this effect needed to be examined. Twelve weeks after C2 hemisection, two new groups of animals were injected with ChABC (Fig. 3a) and received 1 week of either IH conditioning or air. Following the end of treatment, the animals were housed for a further 3 or 6 months before end point recordings were conducted (Supplementary Table 1, group 3).

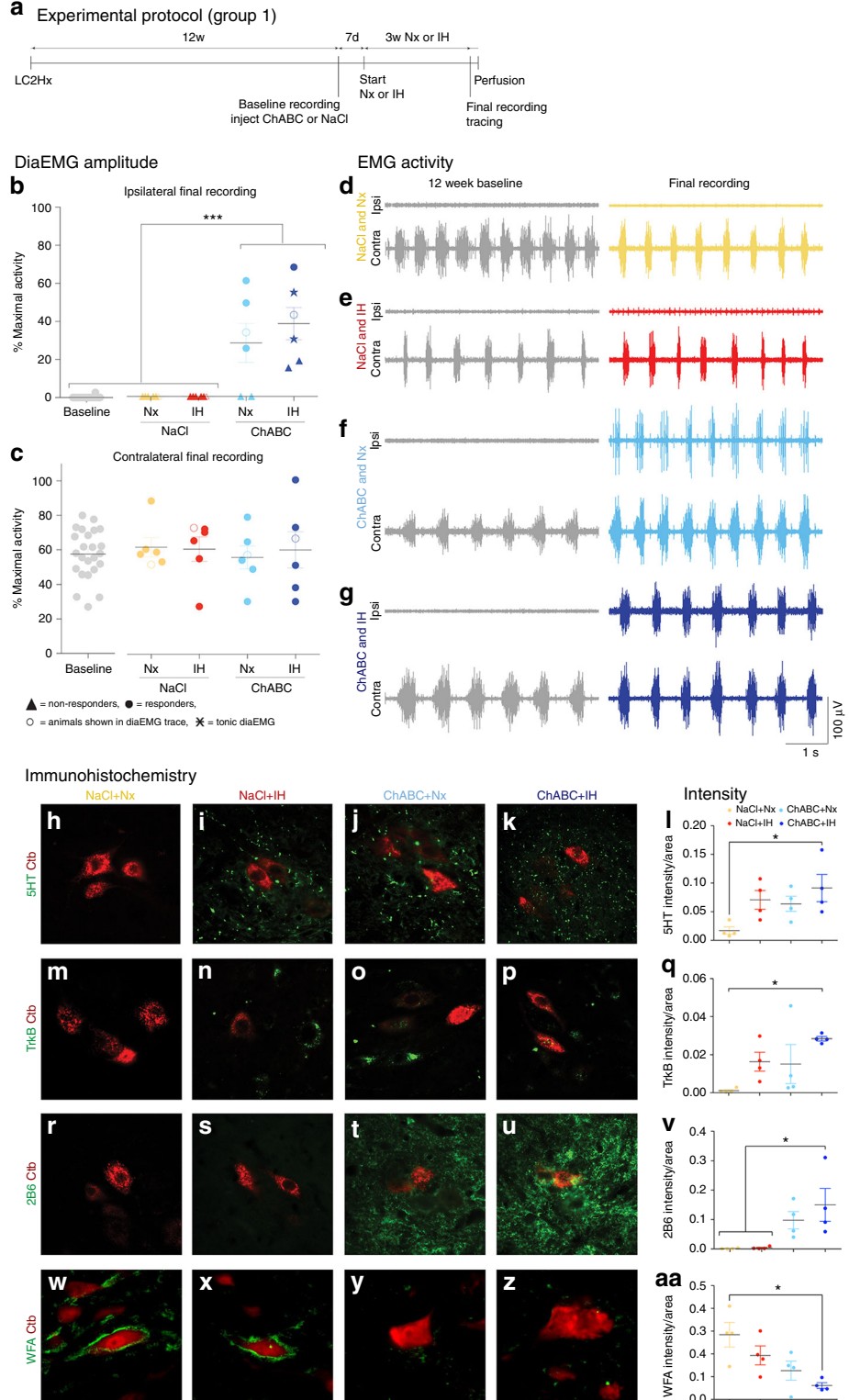

**Fig. 1** PNN breakdown restores ipsilateral hemidiaphragm function 12 weeks after cervical SCI. **a** Experimental protocol. **b**, **c** Average amplitude of **b** ipsilateral and **c** contralateral diaEMG. Filled circles = responders; unfilled circles = animals in **d**–**g**; triangles = non-responders; stars = tonic activity shown. **d**–**g** Representative EMG recordings at baseline and following the 4 weeks of treatment for all groups. Data panels presented from the same animal. For **b**–**g** $n = 6$ per group, baseline $n = 24$. **h**–**aa** Immunohistochemistry and intensity readings at the ipsilateral C4 PMP utilizing Ctb (red) and either (green) **h**–**l** 5HT, **m**–**q** TrkB, **r**–**v** 2B6, or **w**–**aa** WFA ($n = 4$ per group). For all EMG or graph panels, treatment groups = baseline (grey); saline+air (yellow); saline+IH (red); ChABC+air (light blue); ChABC+IH (blue). *$p < 0.05$, **$p < 0.01$ and ***$p < 0.001$. If no post-hoc result is shown, comparison was not-significant. Scale bar = 50 µm. For all panels: values represent mean ± SEM

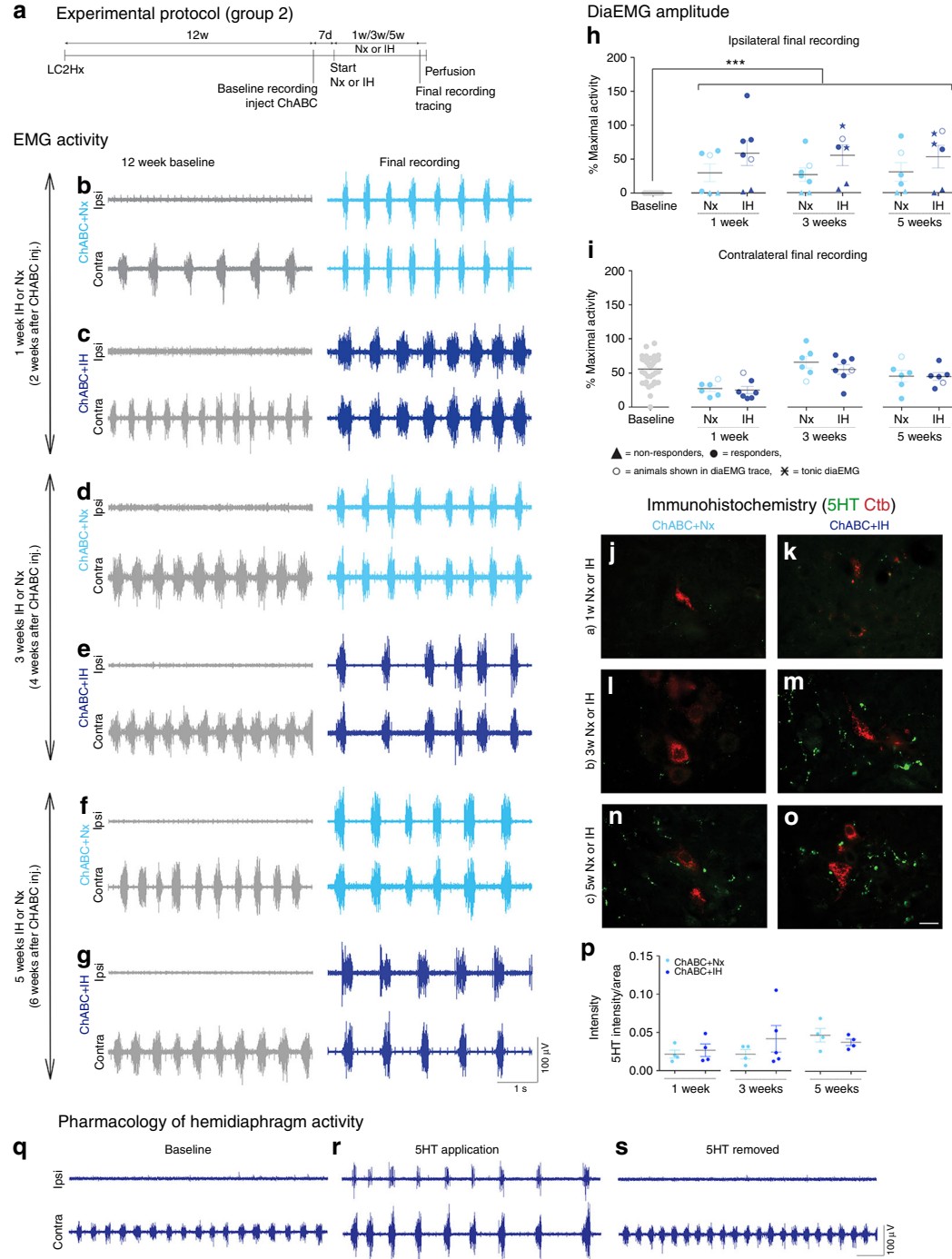

**Fig. 2** Matrix remodelling recovers respiratory activity at multiple time points. **a** Experimental protocol. **b–g** DiaEMG recordings at baseline and following treatment. Animals received ChABC and air (Nx) or IH conditioning for **b**, **c** 1 week; **d**, **e** 3 weeks; or **f**, **g** 5 weeks. Data panels presented from the same animal ($n = 6/7$). **h**, **i** Average amplitude of **h** ipsilateral and **i** contralateral diaEMG. Filled circles = responders; unfilled circles = animals in **b–g**; triangles = non-responders; stars = tonic activity shown. For **b–i**, $n = 6/7$ per group, baseline $n = 38$. **j–p** Immunohistochemistry and intensity readings at the ipsilateral C4 PMP utilizing Ctb (red) and 5HT (green; $n = 4/5$ per group). **q–s** Activity is present in the ipsilateral hemidiaphragm following application of exogenous 5HT ($n = 9$). For all EMG or graph panels, treatment groups = baseline (grey); ChABC+air (light blue); ChABC+IH (blue). ***$p < 0.001$. If no post-hoc result is shown, comparison was not-significant. Scale bar = 50 μm. For all panels: values represent mean ± SEM

Three months after treatment end, 2/3 of the animals in each group demonstrated strong synchronised ipsilateral diaphragm activity, identical to that present immediately after treatment end (ANOVA, $p < 0.0001$, $F = 14.4$; Fig. 3b, c, f). The amplitude of this response was statistically similar to that of the contralateral hemidiaphragm, operating now at ~60% of maximal in both

groups. Importantly, these recuperative effects were maintained in the animals assessed 6 months after the end of treatment (Fig. 3b–e, g). DiaEMGs revealed enhanced synchronised ipsilateral diaphragm activity (ANOVA, $p < 0.0001$, $F = 16.38$) operating at approximately the same % of maximal output as an uninjured animal. Also, critically, 100% of animals assessed

6 months after the end of treatment showed restoration in the ipsilateral hemidiaphragm (Fig. 3g). This would suggest that the treatment triggered recovery processes continued long after its conclusion, and that some animals just take longer to show the effects produced by ECM removal.

The robustness of ChABC-mediated ECM breakdown to engender recovery throughout the respiratory motor system was seen again through diaNMJ analyses (Supplementary Figure 4j-m). The amount of intact, denervated and reinnervated diaNMJs at 6 months post-treatment was the same as that produced immediately after treatment end (Supplementary Figure 4e-m) with no statistical differences between the air and IH groups (ANOVA). Interestingly, the ventilatory response showed a substantial increase at 3 and 6 months after treatment end (Supplementary Figure 5d-f). This effect was mediated by both tidal volume and frequency. The reason for this increase in response in both treatment groups between 14 and 98 days post-treatment should be further explored, however, it was persistent lasting up to 183 days post-treatment (ANOVA, HVR $p < 0.0001$ for all measures; HCVR $p < 0.05$ for all measures).

The long-lasting effect caused by ChABC treatment may be due to the reformation of the PNNs at the level of the ipsilateral and contralateral PMPs (Fig. 3l–n; Supplementary Figure 6e-g; ANOVA, ipsilateral $p = 0.3505$, $F = 28.67$; contralateral $p = 0.1678$, $F = 7.8$), that serve to stabilize the functional changes within the spinal cord. The lack of ChABC activity at this time could be seen through the substantial reduction in stub antigen (2B6) staining (Fig. 3j, k, n; Supplementary Figure 6c-d,g; ANOVA, ipsilateral $p = 0.7483$, $F = 3.494$; contralateral $p = 0.3979$, $F = 4.188$). However, levels of 5HT positive fibres within the ipsilateral (Fig. 3h, i, n) and contralateral (Supplementary Figure 6a-b,g) PMPs remained relatively high (ANOVA, ipsilateral $p = 0.0016$, $F = 3.294$; contralateral $p = 0.0053$, $F = 2.875$).

**Return of function in multiple motor systems**. While able to ambulate, feed, drink, etc., C2 hemisected animals exhibit deficits in forelimb behaviour. This includes reduced gross and fine movements of the ipsilateral limb (Supplementary Figure 7b-g). Despite our treatment application not being optimised for effects upon the forelimb (specifically in the spinal placement of ChABC at C4), the strategy none-the-less elicited improvements in upper arm function, likely mediated via enzyme diffusion caudally. Indeed, we show enzyme activity both ipsilateral and contralateral to the original injury at C6 of the spinal cord in close proximity to the rubrospinal tract (Supplementary Figure 7h-j; ANOVA, group 4 = ipsi: $p < 0.001$, $F(3,11) = 13.94$; contra: $p = 0.0146$, $F(3,11) = 6.64$; group 3 = ipsi: $p = 0.6951$; contra: $p = 0.755$). In animals injured for 3 months prior to treatment with ChABC and either 5 weeks of IH conditioning or air (Supplementary Figure 7a), the forelimb asymmetry test showed significant differences between treatment groups (ANOVA, $p < 0.0001$, $F(2,24) = 11.29$; Supplementary Table 1, groups 3a+b, 4) over time (ANOVA, $p < 0.0001$, $F(6,144) = 28.46$). Animals receiving the enzyme more readily used both limbs to move about and explore their environment (~70% compared to ~30% in control animals; Supplementary Figure 7b-d). This effect was sustained up to 3 months following the end of treatment (ANOVA, drug: $p = 0.1942$, $F(1,12) = 1.89$; time: $p < 0.0001$, $F(14,168) = 6.27$). Noteworthy was the trending effect that IH alone had upon forelimb function (Supplementary Figure 7). While not significantly different from controls, this conditioning treatment was sufficient to marginally increase the use of both forelimbs to ~50%. These data support the work in rodent and human subjects showing that IH can improve loco-motor behaviour[16,20]. Importantly, the ChABC+IH-treated

animals also showed significant differences from controls in ipsilateral forelimb behaviour during the grooming test (Supplementary Figure 7e-g). Treatment affected forelimb function (ANOVA, $p < 0.0001$, $F(3,24) = 11.87$) over time (ANOVA, $p < 0.0001$, $F(6,144) = 6.36$) with the ChABC+IH-treated animals showing a greater functional range. This improvement was sustained up to 3 months following the end of treatment (ANOVA, drug: $p = 0.8366$, $F(1,12) = 0.04$; time $p < 0.0001$, $F(14,168) = 4.93$). Restoration of gross upper forelimb behaviour would suggest possible sprouting of the rubrospinal tract[7] and that our treatment strategy is able to functionally modulate multiple motor systems simultaneously.

**Plasticity can cause tonic hemi-diaphragm firing**. The combined ChABC+IH-induced restoration of respiratory function did not always yield a normally patterned respiratory diaEMG response (Fig. 4a, b). 33.3% (or 50% of those with hemidiaphragm recovery) of the animals that had received ChABC at 3 months post-SCI and either 3 or 5 weeks (but not 1 week) of IH conditioning (star data points, Figs. 1b, 2h, Supplementary Table 1, groups 1d, 2c+f, 3b, 4d) presented tonic firing patterns only in the ipsilateral hemidiaphragm (Figs. 4a, b, 5; Supplementary Movie 5). This occurred either continuously or as waves of firing (Supplementary Figure 8a-b). While the severity of this activity varied (Fig. 5), cycle-triggered averaging (CTA) showed it to be somewhat structured. Motor units increased firing in synchrony with the contralateral hemidiaphragm to initiate inspiration and expiration (Figs. 4c–f, 5). Importantly, this unique activity diminished by the 5th week of IH treatment (Figs. 4a–f, 5), correlating with PNN reappearance (Supplementary Figure 3c). Indeed, 3 months after the end of treatment some aberrant single motor unit firing was evident in two ChABC+IH-treated animals (Supplementary Figure 6h). Tonic activity was not present in any animals 6 months after the end of treatment (Supplementary Figure 6i).

**Tonic activity: 5HT and receptor imbalances**. The mechanism underlying tonic respiratory motoneuron activity had to be revealed to understand its effect upon ventilation. We divided our immunohistochemical analysis of the ChABC+IH treatment groups into non-responders, normally patterned responders and those that showed tonic motoneuron firing (Fig. 4g). With 3 weeks of IH, animals that displayed tonic diaphragm activity demonstrated especially high levels of 5HT (Fig. 4g, i, k; Supplementary Table 1, groups 1d, 2c). At 5 weeks of IH treatment, the amount of 5HT in the ipsilateral PMP had diminished, correlating with the reduction of tonic motor unit firing in the diaphragm (Fig. 4b, g, i, k; Supplementary Table 1, groups 2f, 4d). These trends were mirrored in the contralateral PMPs (Supplementary Figure 9o).

When divided by treatment response, the constitutively active serotonergic receptors 5HT2c/a increased within the ipsilateral motor pool in animals that responded to ChABC+IH (Fig. 4h; Supplementary Figure 9a-n). This occurred more modestly in the contralateral PMP and in response to ChABC treatment alone (Supplementary Figure 9o-q). These receptors are upregulated following SCI to augment the excitability of spinal motor neurons without being dependent upon 5HT itself[26]. As such, they may play a partial role in causing the restoration of function. However, animals that displayed tonic motor unit firing ipsilateral to the injury did not have unusually high numbers of 5HT2c/a receptors (Fig. 4k, l).

What might be the effect of further increasing serotonergic activity in animals that responded normally to the combination treatment or blocking 5HT function in those where respiration

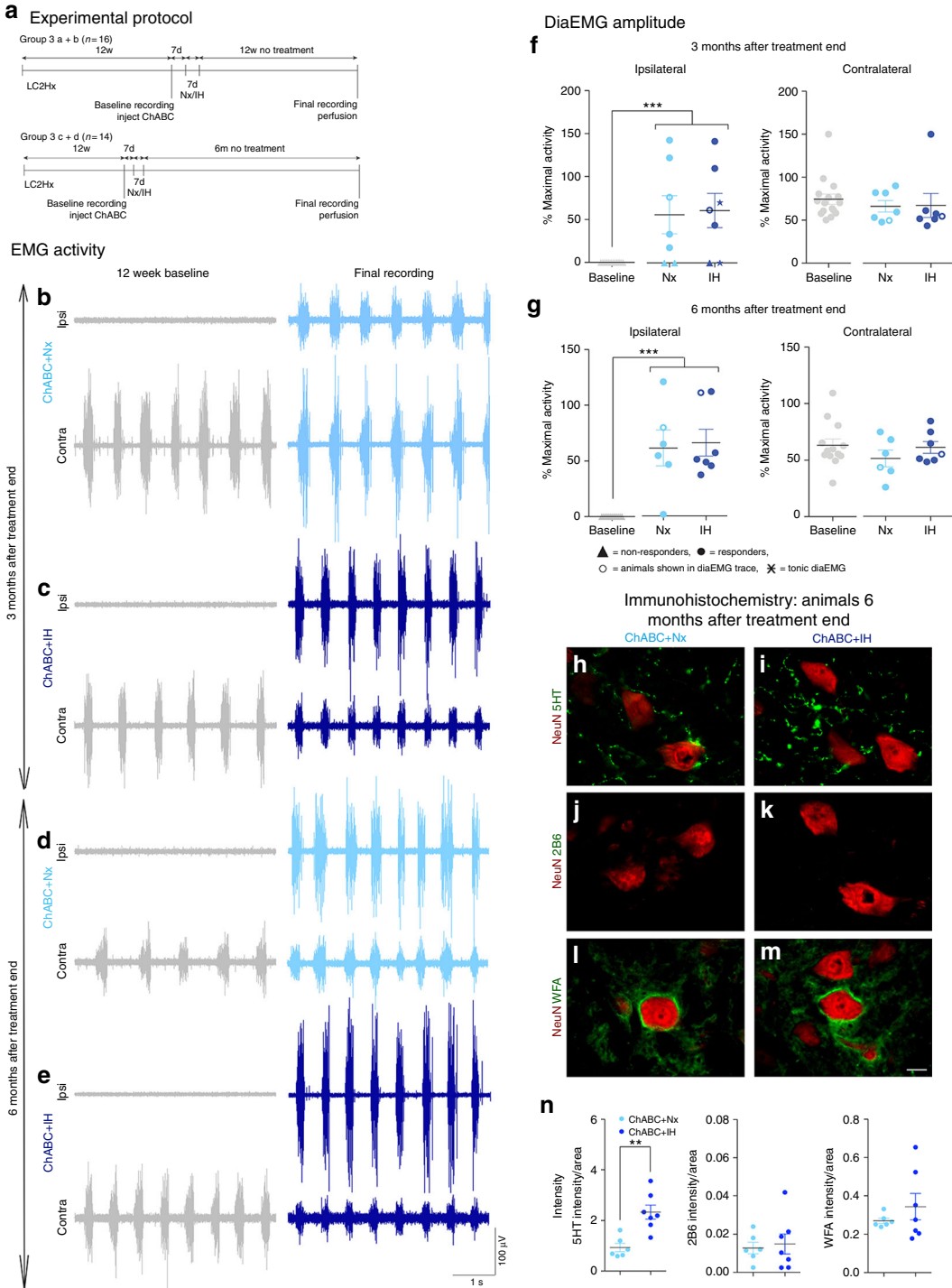

**Fig. 3** Respiratory recovery is maintained up to 6 months after PNN digestion. **a** Experimental protocol. **b–e** DiaEMG recordings at baseline and after treatment. Animals received ChABC and air (Nx) or IH conditioning and were housed for **b**, **c** 3 months or **d**, **e** 6 months without treatment. Data panels presented from the same animal. **f**, **g** Average amplitude of **f** 3 month and **g** 6 month animals diaEMG. Filled circles = responders; unfilled circles = animals in **b–e**; triangles = non-responders; stars = tonic activity shown. For **b–g** $n = 6/7$ per group, baseline $n = 13/14$. **h–n** Immunohistochemistry and intensity readings at the ipsilateral C4 PMP utilizing NeuN (red) and either (green) **h**, **i** 5HT; **j**, **k** 2B6; or **l**, **m** WFA 6 months after treatment ($n = 6/8$ per group). For all EMG or graph panels, treatment groups = baseline (grey); ChABC+air (light blue); ChABC+IH (blue). **\*\***$p < 0.01$ and **\*\*\***$p < 0.001$. If no post-hoc result is shown, comparison was not-significant. Scale bar = 50 μm. For all panels: values represent mean ± SEM

was tonic? We applied 5HT to a subset of ChABC+IH-treated animals that displayed recovery of synchronized ipsilateral hemidiaphragm activity (Fig. 4m; $n = 7$; 5 systemic applications, 2 direct injections). Following application, in all animals the ipsilateral hemidiaphragm rapidly developed tonic motor unit

firing that did not dissipate until the neuromodulator had likely been metabolized. Further, in all animals that exhibited tonic ipsilateral activity ($n = 6$; 4 systemic applications, 2 direct injections), application of a 5HT2c/a antagonist blocked the abnormal motoneuron firing revealing respiratory activity

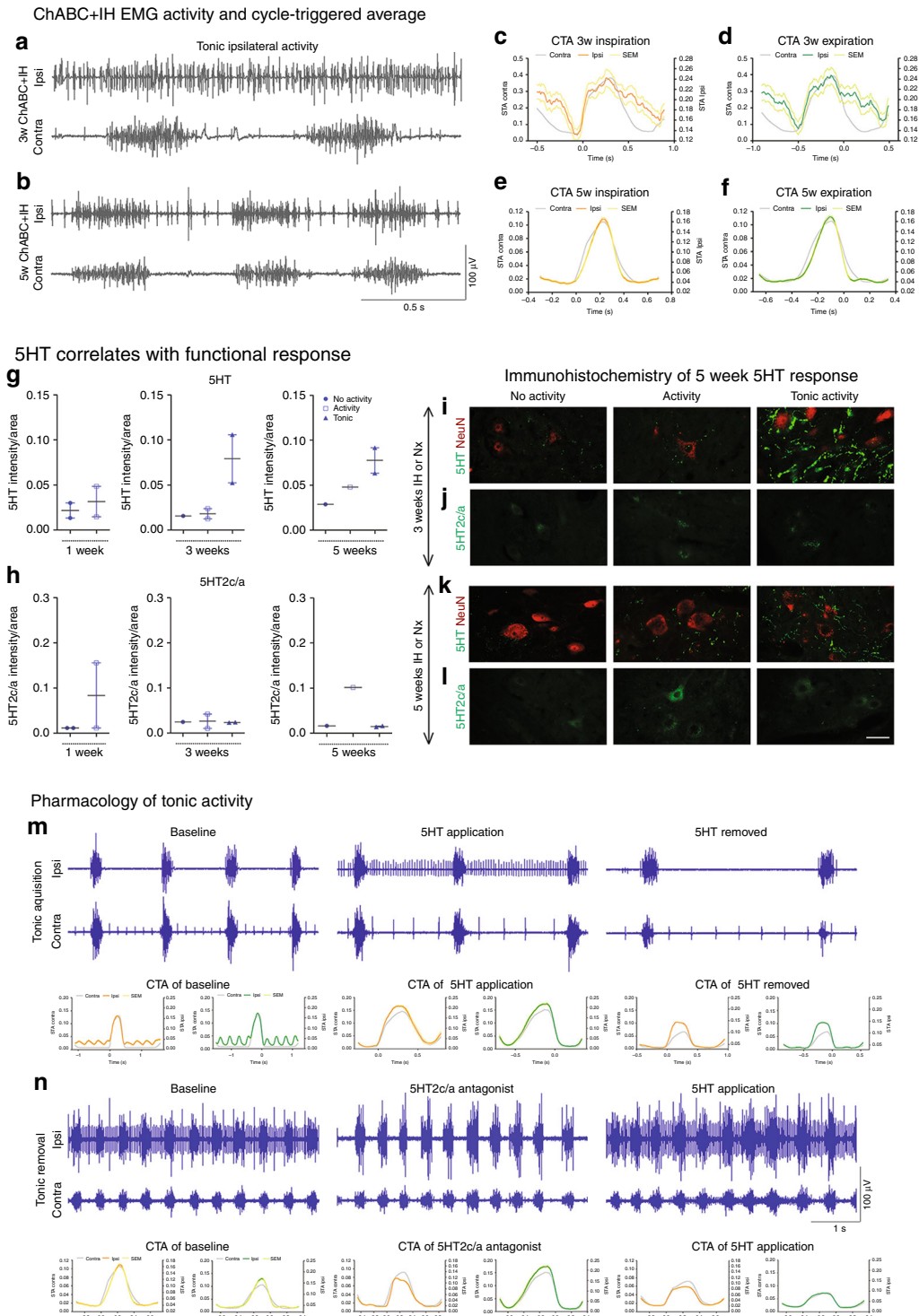

**Fig. 4** Excessive 5HT and receptor imbalance causes ipsilateral tonic activity. **a–f** Tonic diaEMG activity following **a** 3 or **b** 5 weeks conditioning. **c–f** Representative CTAs of ipsilateral (coloured) and contralateral (grey) activity for mean inspiration (orange) and expiration (green) where ±SEM is shown for both inspiration (yellow) and expiration (light green). IH for **c**, **d** 3 or **e**, **f** 5 weeks ($n = 6$, see Fig. 5). **g–l** Intensity readings for **g** 5HT, **h** 5HT2c/a and **l** representative images at the ipsilateral C4 PMP utilizing **i**+**k** NeuN (red) and 5HT (green) or **j**+**l** 5HT2c/a (green). Graphs show mean ± SEM. Data grouped on ipsilateral hemidiaphragm activity: non-responders (circle; $n = 1/2$), normal activity (square; $n = 1/2$), tonic (triangle; $n = 2$). **m**, **n** Tonic activity following **m** 5HT ($n = 7$) and **n** 5HT2c/a antagonist ($n = 6$) application. CTAs of ipsilateral (coloured) and contralateral (grey) activity for mean inspiration (orange) and expiration (green) where ±SEM is shown for both inspiration (yellow) and expiration (light green) of pharmacologically induced and removed tonic activity. Panels recorded in the same animal. For all EMG or graph panels: animals were treated with ChABC+IH (blue)

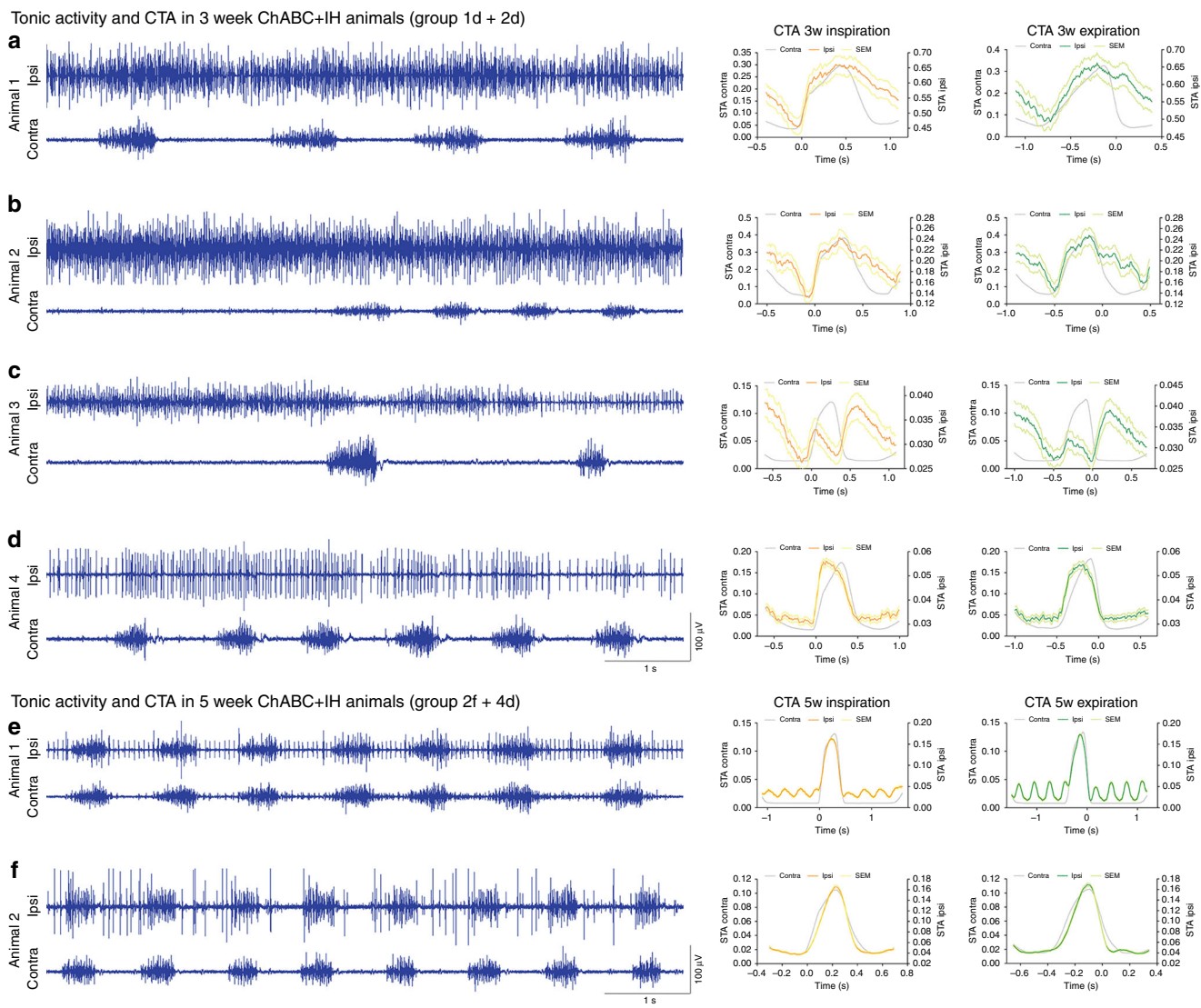

**Fig. 5** Representative diaEMG traces of ChABC+IH (blue) animals that displayed tonic firing of the ipsilateral hemidiaphragm ($n = 6$). **a–d** Tonic activity exhibited in animals with 3 weeks of conditioning. CTAs generated with triggers at the onset of inspiration and expiration. Panels (**b**, **c**) demonstrate that the onset of tonic ipsilateral hemidiaphragm firing can reduce the firing frequency of the contralateral hemidiaphragm. **e**, **f** Tonic activity exhibited in animals with 5 weeks of conditioning. CTAs reveal inhibition of tonic activity before the onset of inspiration and expiration. CTA on panel (**e**) shows the effect on cardiovascular activity on the trace. Panels recorded in the same animal. For all panels: animals were treated with ChABC+IH (blue EMG traces); CTAs shown of ipsilateral (coloured) and contralateral (grey) activity for mean inspiration (orange) and expiration (green) where ±SEM is shown for both inspiration (yellow) and expiration (light green)

synchronized to the contralateral hemidiaphragm (Fig. 4n). Subsequent application of 5HT caused the reappearance of tonic ipsilateral motor unit firing, with the same abnormal patterning as seen initially (Fig. 4n). CTA analysis revealed patterning within this abnormal activity. Collectively these data show that, in the presence of excessive amounts of 5HT, even a moderate density of the constitutively active 5HT2c/a receptors, can cause tonic excitability of the ipsilateral phrenic motor neurons.

**Breathing restoration 1.5 years after paralysis**. To fully elucidate the potential for our combination therapy to stimulate respiratory motor recovery at protracted time periods following spinal trauma, we treated animals with either vehicle or ChABC that had been paralysed by a complete C2 hemisection 1.5 years earlier (see Supplementary Table 1, group 5 for discussion of group allocation).

Following 1.5 years of complete hemidiaphragm paralysis (Fig. 6b, c), some activity returned to the ipsilateral hemidiaphragm just 7 days after ECM breakdown by ChABC in 40% of the animals (Fig. 6c, d). Application of the vehicle did not cause functional diaphragm recovery and there was no evidence of a compensatory increase in the contralateral diaEMG (Fig. 6b, d; Supplementary Figure 10a). Enzyme activity was confirmed through WFA and 2B6 staining (Supplementary Figure 10h-o,y-af). This rapid restoration in activity was surprising considering the diaphragm starts to atrophy within 24 h following C2 hemisection[27], and may explain why the amplitude of activity was initially small. One week later, 60% of the animals responded to ChABC treatment demonstrating strong synchronized dia-EMG activity (Fig. 6e–g). With an additional 2 weeks, 100% of the animals had responded to the treatment (ANOVA, $p = 0.012$, $F = 2.706$). Animals treated with ChABC and either air or IH breathed during eupnea at ~45% of their maximal capacity, values

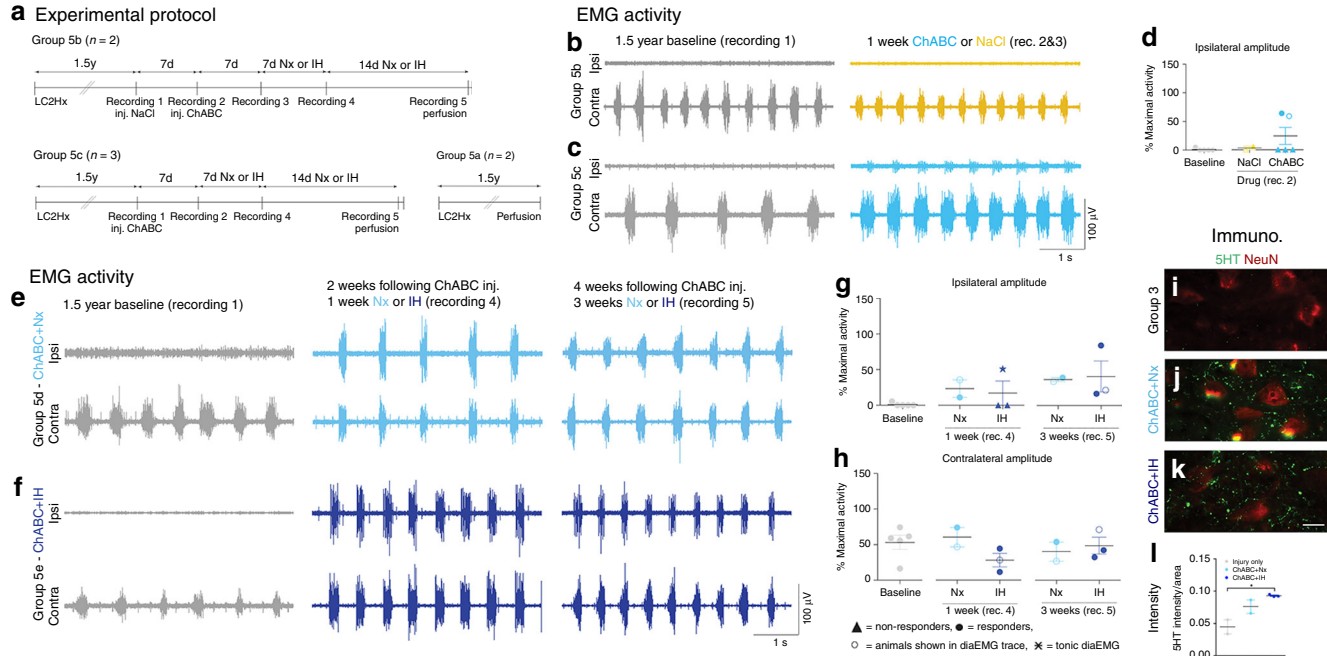

**Fig. 6** PNN digestion recovers respiratory activity a year and a half after spinal injury. **a** Experimental protocol. **b–d** DiaEMG at baseline (grey; $n = 5$) and following either **b** NaCl (yellow; $n = 2$) or **c** ChABC (light blue; $n = 3$). **d** Average amplitude of ipsilateral diaEMG. **e–h** DiaEMG at baseline (grey; $n = 5$) and following ChABC and **e** air (light blue; $n = 2$) or **f** IH (blue; $n = 3$). **g, h** Average amplitude of **g** ipsilateral and **h** contralateral diaEMG. **i–l** Immunohistochemistry and intensity readings at the ipsilateral C4 PMP utilizing NeuN and 5HT for **i** injury only, ChABC and **j** air or **k** IH. Scale bar = 50 μm ($n = 2/3$ per group). For all EMG or graph panels, treatment groups = baseline (grey); saline+air (yellow); ChABC+air (light blue); ChABC+IH (blue). For **d, g, h**: filled circles = responders; unfilled circles = animals in **b, c, e, f**; triangles = non-responders; stars = tonic activity shown. If no post-hoc result is shown, comparison was not-significant. For all panels: values represent mean ± SEM

not statistically different from the contralateral hemidiaphragm (Fig. 6h). A meagre amount of tonic activity was demonstrated in one ChABC+IH-treated animal following 1 week of conditioning (Supplementary Figure 8c). Interestingly, this abnormal response vanished with 2 weeks further treatment, confirming that tonic activity is removed over time.

This noteworthy restoration of respiratory function 1.5 years following SCI correlates with an increase in 5HT at the PMP ipsilateral to the injury (Fig. 6i–l). Similarly, it is associated with increases in the 5HT2c/a and TrkB receptors in both the ipsilateral and contralateral PMPs (Supplementary Figure 10d-k, t-ab). Further, the frequency of breaths was not statistically different between all treatment conditions (Supplementary Figure 10b-c; ANOVA, $p = 5.611$, $F = 0.8357$).

### Discussion

We show that chondroitinase digestion of CSPGs within the ECM following a single, minimally invasive local dose in the vicinity of the PMP can recover respiratory motor function up to a near lifetime of diaphragm paralysis. This is the first illustration that persistent, complete functional restoration in any motor system is possible well over a year following SCI. Further, the rapid restoration of function is superior to anything achieved at acute or sub-chronic stages after injury using similar techniques, suggesting that this plasticity inducing or revealing treatment may be best applied not within days or weeks (as is typically done) but many months after injury to maximise its effect. Moreover, we show that the recovered behaviour is maintained up to 6 months after the end of treatment in numerous effectors within the respiratory motor system including ventilatory function, muscle activity and the physiological state of muscle–nerve junctions.

Our data adds to and refines the current thinking concerning the two treatment strategies we have used in combination. Firstly, we show that in the absence of spontaneous recovery, IH does little to recover respiratory motor function at long chronic time points. This suggests that for IH to be an effective form of rehabilitation, a motor system needs to be reasonably active rather than completely paralysed. Thus, IH may be able to augment existing function, rather than recover it[20]. Further, it has previously been shown that ChABC treatment can open a window of opportunity for the addition of task-specific training to have a meaningful functional benefit on the locomotor system[14]. However, our data suggest that ChABC alone, especially at long protracted times after SCI, is both necessary and sufficient to recover substantial function at all levels of the respiratory motor system. The question remains as to why the long paralysed and atrophied hemidiaphragm can recover so quickly and fully without extensive respiratory rehabilitation? It is likely that, while the ipsilateral motor pathways are not directly activated, the constant passive stretching of the paralysed hemidiaphragm (see Supplementary Movies 1 and 2) may act as a form of rehabilitation aiding protein synthesis as well as the upregulation of neurotrophic factors thereby minimising the loss of muscle mass and NMJs over time[28,29]. As such, the muscles may be primed for recovery, requiring only the reformation of sufficient connections within the motor circuitry.

What is the anatomical substrate that mediates such robust recovery? As regeneration through the site of injury did not occur, it is possible that it involves the activation of latent pathways[17] or treatment-induced plasticity. Motor circuits deprived of supraspinal input undergo substantial remodelling continuing long after insult[30,31]. In our complete hemisection model, sprouts[32] of latent projections arising from the contralateral

respiratory groups (with axons that cross the midline below the level of injury or intervening interneurons with similar decussating properties[17,33]) may be slowly forming new, potentially active synapses over time that could be beneficial, harmless or perhaps even detrimental. It is possible that this involves recruitment of midline crossing V2a interneurons, which have been shown to be critical for respiratory function and recovery following high cervical SCI in both mice[34,35] and rats[36]. Indeed, using an ex vivo prep we have recently shown that latent glutamatergic spinal interneuronal networks in adult mice and rats can, indeed, be harnessed to restore diaphragm function even after debilitating injury[37]. Moreover, IH has recently been implicated in the augmentation of propriospinal networks due to its ability to increase the coupling of mid-cervical interneurons with phrenic motoneurons[38]. We are currently using rat and transgenic mouse models to address the precise mechanism through which the recovery we describe has occurred and the extent to which mid-cervical interneurons are involved[38]. However, importantly, we demonstrate here that the potential activity mediated via these newly generated connections is completely curtailed by CSPGs in the PNN or surrounding ECM. This demonstrates a novel way in which CSPGs do not just limit growth and plasticity, but are involved in masking, stabilizing and controlling activity within a system. The mechanism by which CSPGs inhibit transmission still needs to be elucidated[39] but it is conceivable that they do so by clinging to astrocyte processes that have been shown to wedge themselves between synapses on phrenic motor neurons[40]. We demonstrate that ChABC acts to digest this matrix and, in conjunction with a renewed supply of 5-HT, likely rapidly unmasks the functional potential of the sprouting that had already occurred. However, we do not rule out the possibility that matrix digestion might stimulate further neuronal sprouting[11] to enhance the activity of these plastic fibres[41]. The persistence of functional recovery over 6 months after the end of treatment and reduction in tonic motor neuron firing is likely due to the re-occurrence of the PNN at the level of the PMP and the subsequent incorporation of the anatomical changes engendered through our treatment strategy into the circuitry of the spinal cord.

An additional novelty of the long-delayed recovery of respiratory function is the generation of tonic hemidiaphragm activity. This form of unusual muscle stimulation may be beneficial to muscle strengthening, but detrimental to respiratory function. It appears analogous to debilitating muscle spasms that develop in other systems following SCI that have been shown to be partially dependent on spontaneously active 5-HT2C receptors[26,42,43]. We have determined that a major part of the mechanism behind this unique activity involves the raphe-spinal system. Excesses in the amount of 5HT, brought about only by the IH/enzyme combination, could result in large depolarisations and enhanced membrane resistance in the phrenic motoneurons acting, in turn, to increase end-expiratory lung volume facilitating ventilation[44,45]. Alternatively, the ipsilateral spasm-like activity may be detrimental to ventilation by compromising lung function. The potential for untoward activity could explain why the CNS acts to suppress potentially aberrant motor system regeneration and plasticity following trauma through proteoglycan upregulation within the ECM and PNN. The reduction in tonic diaphragm activity over time during and after the end of treatment could similarly be considered comparable to the connections that form during the development of motor circuits that are refined with activity[46,47]. Indeed, increased 5HT is known to play a role in the development, stabilization and maturation of the respiratory control system[42,48] and our treatment strategy may be recapitulating this process.

Our data illustrate the relative ease with which an essential motor system can regain functionality months to years after severe hemiparalysis. The mechanism of this repair induced in our model may have developmental and locomotor correlates following chronic spinal trauma in other systems with re-crossed projections such as the corticospinal, rubrospinal or propriospinal tracts[11,12,16]. Indeed, we demonstrate that our simple treatment strategy has simultaneous functional effects upon forelimb motor function. These data suggest that the fundamental principles driving persistent restoration of function after chronic trauma is not limited to the respiratory motor system and may operate through parallel mechanisms.

## Methods

**Ethical declaration and animal husbandry.** All experiments were approved by the Institutional Care and Use Committee at Case Western Reserve University, Cleveland. Animals were housed in groups of three, exposed to a normal dark–light cycle with free access to food, water and environmental enrichment ad libitum. The health and welfare of the animals was monitored by the study investigators and veterinary staff at Case Western Reserve University.

**Statistical assessment.** For statistical analysis, divergences were considered significant if $p < 0.05$. Data are presented as mean ± SEM. Power analysis was conducted prior to all experiments to ensure $n$ numbers were sufficient to yield reliable data and were within the 88–96 percentiles. In instances where power was not sufficient (due to experimental groups being sub-divided based on experimental outcome), analysis was not conducted. However, in this instance, the statistical analysis of total group outcomes was reported. At the time of recording, processing and analysis of all experiments and assessments, investigators were blind as to the treatment group of each animal. Data were subjected to the Shapiro–Wilk test for normalcy prior to analysis to ensure a normal distribution. The functional and behavioural recordings from every animal in each group were analysed without exclusion based on the outcome. Division of animals into specific groups for the experiments described has been outlined in Supplementary Table 1.

**Surgical procedures: injury and spinal injections.** Adult female Sprague Dawley rats (200–250 g; Harlan Laboratories Inc., Indianapolis, IN, USA) were anaesthetized with a ketamine/xylazine cocktail (70 mg kg$^{-1}$/7 mg kg$^{-1}$) through intraperitoneal (i.p.) injection. Once the surgical plane of anaesthetic depth had been reached the animals were prepared for surgery by shaving and cleansing the dorsal neck with Betadine and 70% ethyl alcohol and analgesics were administered through subcutaneous (s.c.) injection of Carprofen (5 mg kg$^{-1}$) and i.m. injections of 0.002% bupivacaine hydrochloride along the site of incision. Body temperature was maintained throughout the surgery at 37 ± 1 °C. A dorsal midline incision ~3 cm in length was made over the cervical region and the skin and paravertebral muscles were retracted. A laminectomy was performed over C2 and C3 exposing the rostral spinal cord. Using a 21G needle (with the tip positioned towards the mid-line), a left lateral durotomy and hemisection were performed caudal to the C2 dorsal roots. The incision was made to the ventral lamina surface. This process was repeated five times and extended from the midline to the most lateral extent of the spinal cord. The anatomical completeness of the injury was confirmed through microscopy (Supplementary Figure 1c). The muscle layers were sutured together (3-0 vicryl) and the skin closed using wound clips. The animals were given Buprenorphine (30 μg kg$^{-1}$) and saline subcutaneously. This method of analgesia and hydration was maintained up to 5 days post-surgery along with nutritional support should the animals weight have dropped more than 5% of that pre-injury.

One week post-injury, the animals (housed in triplicate) were fully motile and able to move, eat, drink, etc., with autonomy. Of the total number of animals used in this study (see Supplementary Table 1), 1 stopped breathing immediately following the injury and could not be recovered with manual ventilation. Of the remaining rats, no animal kept for 3 months post-injury before treatment application died due to complications relating to the initial spinal injury or the experimental protocol. Of the animals kept for 1.5 years following injury, 3 of the animals were sacrificed ~1-year post-injury due to complications of aging unrelated to the initial trauma (natural death and the development of mammary tumours). Approximately 5–8 weeks following injury, 10 animals showed increased biting of their right hind limb. This was typically resolved through supplying all animals with additional environmental enrichment. Four animals had the tips of their right hind limbs removed following veterinary advice which prevented further complications.

A number of laboratories have demonstrated varying degrees of spontaneous hemidiaphragm recovery following this type of injury[49,50]. The cause of this improvement may be due to the activation of decussating latent or recruited projections, surviving ventral medial tissue at the point of injury, the imprecise segmental location of the trauma, the effect of specific anaesthetics on respiratory output at the time of recording, or sex and strain differences[17,51,52]. Our model results in complete and persistent ipsilateral hemidiaphragm paralysis.

Subsequently, our model reproduces the effects of severe human cervical SCI, resulting in permanent hemidiaphragm paralysis[53] and provides a solid baseline from which to examine respiratory recovery. Furthermore, the majority of human SCIs demonstrate sparing in a variable portion of descending axons. The complete hemisection represents such a partial lesion with the benefit of generating reproducible sparing from the contralateral side which, in turn, may potentially be recruited to reactivate denervated phrenic motor neurons.

Since the recovery we have obtained is likely dependent upon the sprouting of a reproducibly spared population of contralateral axons, one wonders whether this strategy may be translated to more severe and variable human contusive injuries that affect the cord bilaterally. Depending upon the amount of anatomical sparing following such injuries, a degree of spontaneous diaphragm recovery can occur[54]. Thus, in patients with some remaining descending supraspinal projections to the cord, perhaps minimally sufficient to allow them to be partially respirator independent, it is conceivable that this approach could also be successful. However, following more complete cervical injuries, use of the enzyme alone is unlikely to have significant benefit, without the construction of a regeneration promoting bridge[10].

Similar to acute injury studies[10], the chronically injured animals received an injection of either ChABC (Seikagaku; 20 U mL$^{-1}$) or a saline vehicle control. Investigators were blind as to which drug each animal received. Animals were anaesthetized via i.p. injection with the ketamine/xylazine cocktail. Once the animals had reached a surgical plane of anaesthesia, the dorsal neck was shaved and cleansed with Betadine and 70% ethyl alcohol. Carprofen was administered s.c. and bupivacaine hydrochloride i.m. along the site of incision. Body temperature was maintained through the procedure at $37 \pm 1$ °C. A dorsal midline incision opened the skin and paravertebral muscles again and they were retracted. A laminectomy was performed over C4 and C5 and the dura cut. A pulled pipette attached to a Nanoject II (Drummond Scientific Company) was stereotaxically placed at the level of the phrenic nucleus (1.1 mm left of midline and 1.6 mm ventral from the spinal cord dorsal surface). After placement and a 5-min rest period, 250 nL of drug or vehicle was injected into the spinal cord. Once the pipette was removed, the muscle layers were sutured together and the skin closed with wound clips. Buprenorphine and saline were given s.c. with the analgesics maintained for up to 5 days.

**Intermittent hypoxia conditioning**. Acute intermittent hypoxia (AIH) started 1 week following ChABC/saline injection. For 5 days each week animals were placed in a chamber with access to food and water ab librium. The chamber was flushed with air and O$_2$ to attain continuous air (normoxia) or IH (5 min episodes alternating between 10 and 21% O$_2$; representative trace Supplementary Figure 1b). After 10 hypoxic episodes, or the equivalent air (normoxic) duration of 100 min, animals were returned to their cages[16].

**DiaEMG recordings and analysis**. DiaEMG occurred 1 day following the conclusion of IH conditioning. Animals were anaesthetised with a ketamine/xylazine cocktail. Once the animals had reached a surgical plane of anaesthesia, the abdominal region was shaved and cleansed with Betadine and 70% ethyl alcohol. Carprofen was administered s.c. and bupivacaine hydrochloride i.m. along the site of incision. Body temperature was maintained through the surgical procedure at $37 \pm 1$ °C. A 5 cm laparotomy was performed to expose the abdominal surface of the diaphragm. Bipolar platinum electrodes (Grass Technology, Middleton, WI, USA) were placed in the crural region of the left and right hemidiaphragms, dorsal to the anterolateral branch of the inferior phrenic artery. Electrodes were not permanently implanted into the diaphragm due to the inevitable formation of scar tissue at the point of muscle insertion which, over time, diminished the signal output. As such, the same animals had repeated abdominal surgeries to achieve the multiple recordings produced in this study. Activity was amplified (gain 5000×), bandpass filtered (30–3000 Hz; Grass Technology), digitized and recorded using a data acquisition system (CED1401; Spike2; Cambridge Electronic Design). The integrated signal was rectified and smoothed at a time constant of 0.075 s. All recordings were obtained during eupnea prior to a 20-s nasal occlusion (length of occlusion determined due to ethical reasons). The purpose of the nasal occlusion is to induce a reproducible degree of drive and motor recruitment within the system[18]. From this response, the relative activity of hemidiaphragm function can be assessed under eupnea. To ensure of accuracy in reporting, if no activity was demonstrated in the crural region of the hemidiaphragm the sternal and costal regions of the muscle were also assessed for function. Animals which we report as having no EMG activity in the crural diaphragm also showed an absence of activity in the other two regions of the muscle.

For pharmacological assessment of diaEMG activity, drugs were administered both through a tail vein cannula or through intraspinal injection using the same method described above. Animals were dosed with either serotonin (100 mg kg$^{-1}$ i.v.; 120 µg kg$^{-1}$ intraspinal; Sigma) or cinanserin (5 mg kg$^{-1}$ i.v.; 100 mg kg$^{-1}$ intraspinal; Tocris).

If neural tracing was to be performed this occurred following removal of the electrodes. In all animals, the abdominal muscles were sutured together (3-0 vicryl) and the skin closed with wound clips. Buprenorphine and saline were given s.c. with analgesics maintained for up to 3 days.

All EMG recordings were performed with the animal under ketamine–xylazine anaesthesia as is typical within the field[55]. Due to the effect of these anaesthetics, the amplitude and frequency of all our output measures is likely to be reduced compared to that of an unanaesthetised animal[56]. It is doubtful that this has led to us under-reporting the degree of spontaneous recovery in our model, as reported endogenous recovery of ipsilateral nerve and muscle activity following C2Hx has occurred under similar conditions[49,50]. However, this potentially means that the scale of the recovery reported within this manuscript is an underestimation of that actually achieved.

Respiratory frequency, amplitude and bursting activity were determined from each diaEMG trace in 30 s time windows. Amplitude was assessed in relation to the maximal amount achieved for that hemidiaphragm (averaged over three breaths) through the occlusion period to ensure results were not biased by a slight alteration in electrode placement. CTA was performed using CED analysis software (Spike2). CTA facilitated in determining the initiation of inspiration and expiration for animals displaying tonic activity within the EMG trace of the ipsilateral hemidiaphragm. Amplitude was then assessed as stated above. Statistical analysis was performed using a one-way ANOVA with post-hoc Bonferroni (SPSS or GraphPad Prism). Animals from any group or experiment which showed no ipsilateral EMG activity were defined as non-responders.

**Plethysmography recordings and analysis**. Individual animals were placed in Perspex plethysmography chambers (DSI). Chamber temperature was monitored throughout the experiment. Airflow within the chamber was maintained at 1.8 L min$^{-1}$ using flow controllers and hypoxia (10% O$_2$) and hypercapnia (5% CO$_2$) were administered using pre-mixed supplies (AirGas). Levels of O$_2$ and CO$_2$ flowing through the chamber were assessed using gas analysers (ADInstruments, Gas Analyzer ML206). Prior to each recording, the chambers air-tight seal was assessed by injecting a 0.5 mL calibration volume into the chamber and observing the stability of the square pressure change. The same injection volume was later used to calibrate tidal volume (VT) changes in respiratory activity. The animals core body temperature was assessed at the conclusion of each experiment using a rectal thermometer (Physitemp). Through the plethysmography recording (ADInstruments), rates of ventilation (VE) through baseline (normal air), hypoxia and hypercapnia were assessed. Baseline measurements were conducted every 10 min over the course of a 60-min acclimation period, after which the animal received 5 min interval of 10% O$_2$ then 5% CO$_2$. VE was assessed when the chamber was sealed (at the end of each exposure for 30 s) by bypassing airflow to the chamber. Statistical comparisons were made between treatment groups and within plethysmography measurements using one-way or two-way, repeated-measures ANOVA with post-hoc Bonferroni (SPSS).

**Forelimb function assessments**. Behavioural assessments of forelimb function were selected to minimise possible impact on respiratory parameters. As uncontrolled respiratory training was not desired, forelimb function was determined through assessments which involved monitoring behaviour that the animals performed naturally. This included the forelimb asymmetry test and the grooming test, both adapted from Gensel et al.[57]. Baseline values were taken 3 months after injury and prior to ChABC/NaCl administration or Nx/IH conditioning following 5 days of acclimation to the chamber and then weekly following treatment application. Statistical comparisons were made between treatment groups and within behavioural measurements using two-way, repeated-measures ANOVA with post-hoc Bonferroni (SPSS).

For the forelimb asymmetry test, animals were placed in a transparent Perspex cylinder (diameter = 20 cm; height = 48 cm) with food treats placed on a grid at the top. Spontaneous exploratory behaviour was recorded over 5 min. The number of times the animal placed its left, right, or both forepaws on the side of the cylinder was scored under slow motion movie playback for weight supported behaviours. Scoring of limb contact occurred as defined by Gensel et al.[57]. Placement was scored for an individual limb if the contralateral forelimb did not make contact with the cylinder within 0.5 s of the original limb placement. Both limbs were scored if such contact occurred within a 0.5 s time frame. When moving along the side of the cylinder (where both paws change position), again a score for both limbs was given if it occurred within the 0.5 s time frame. However, if one limb remained in its original position, no score was given until the limb was moved.

For the grooming test, and following the forelimb asymmetry test, cool blackcurrant flavoured water was applied to the animal's head and back using a transfer pipette or soft gauze. Grooming activity was recorded over a maximal period of 20 min for both limbs over a minimum of two stereotypical sequences (lasting ~2–2.5 min). Scoring of maximal limb contact occurred as defined by Gensel et al.[57] during slow motion playback.

**Immunohistochemistry, lesion size and tracing**. Animals were anaesthetised through i.p. injection of urethane and perfused intracardially with 250 mL PBS followed by 300 mL of 4% paraformaldehyde (PFA) in PBS. The spinal cord and brain stem were removed and post-fixed in 4% PFA overnight before being cryoprotected in 30% sucrose. The area from C3 to C6 was sectioned on a cryostat (Leica Biosystems) at a thickness of 20 or 25 µm and mounted on SuperFrost coated slides (Fisher Scientific). Tissue was collected utilising the dorsal roots as

landmarks and collected as a function of distance from the lesion site to ensure accuracy and consistency in analysis. Five sections from each segmental level were assessed. The phrenic motor nucleus located in the mediolateral C4 ventral horn is readily recognizable as a tight cluster of large neurons[58].

For immunohistochemistry, mounted sections were washed three times in either TBS or TBST (TBS with 0.1% Triton X-100) depending on the antigen under assessment. Sections were then blocked in 10% normal goat serum and 0.1% BSA in TBS. Sections were incubated overnight in primary antibody at 4 °C. The following day, the sections were washed three times in TBS and incubated in the appropriate secondary antibody or avidin substrate for 2 h at room temperature. After washing three times in TBS, the sections were coverslipped using Fluorogold mounting medium (Invitrogen) and viewed using a fluorescence (Leica) or confocal microscope (Zeiss). 2B6 antibody was purchased from Seikagaku (270432; 1:200), serotonin from Immunostar (20080; 1:1000), 5HT-2c/a from ABcam (ab37293; 1:100), GFAP (G3893; 1:400) and WFA (L1516; 1:50) from Sigma, neuronal nuclei (NeuN) from Millipore (MAB377; 1:400) and tropomyosin receptor kinase B (TrkB) from Biosensis (R121-100; 1:1000). Secondary antibodies (A-11008; A-11012; A-11005; A-11001; A-21042; A-21044; 1:500) and streptavidin Alexa Fluor conjugates (S32354; S32356; 1:500) were from Thermo Fisher Scientific.

For lesion area and volume, six spinal cords were selected for analysis from the total available using a random number generator. Spinal cord lesion area and volume were analysed through iron eriochrome cyanine and 1% cresyl violet staining (Sigma). Following imaging (Leica SCN400 Slide Scanner), the lesioned, white and grey matter areas in each section were determined automatically (Photoshop CC7, Adobe). The lesion volume was assessed using equation $V = \Sigma$ [transplant area × section thickness × 18 (the number of sections in each sampling interval)][59]. Tissue caudal to that used to assess lesion area/volume was used in IHC assessment.

For neuronal tracing, 60 µg of Cholera toxin B subunit conjugated to Alexa-Fluor 488 or 594 (Thermo Fisher Scientific; C34775; C34777; 100% solution) was injected into the left hemidiaphragm using a Hamilton syringe under microscopy 3 days prior to perfusion. As such, the PMP ipsilateral to the injury was localised. Based on NeuN staining, the contralateral PMP in each spinal cord section was also identified. Animals which had undergone over three laparotomies during the course of the experiment were not subjected to tracing due to ethical considerations of an additional surgery. In these instances, the PMP was localised through anatomical location and NeuN staining.

For quantification, the intensity of staining at the defined area of interest was quantified blindly using NIH Image J software (National Institutes of Health; http://rsb.in-fo.nih.gov/ij). The intensity of staining was quantified using monochromatic image and background readings subtracted. Average labelling intensity was compared between the treatment groups using one-way ANOVA with post-hoc Bonferroni (SPSS or GraphPad Prism).

**NMJ staining, quantification and analysis**. Diaphragm NMJs were assessed using the procedure described by Martin et al.[60]. Briefly, the diaphragm was dissected from each animal, stretched, pinned to a Sylgard base (Sigma Aldridge) and cleaned of connective tissue. Motor axons were labelled with anti-neurofilament marker (Covance; SMI-312R; 1:1000) and SV2-s (DSHB; 1:10) and detected with FITC anti-mouse IgG, Fcc1 secondary antibody (Thermo Fisher Scientific; SV2-s; A10530; 1:100). Rhodamine-conjugated alpha-bungarotoxin (Thermo Fisher Scientific; T1175; 1:400) was used to label the post-synaptic acetylcholine receptors (AChR). The muscles were visualised (Leica) and quantified, categorising each NMJ as either (1) intact, (2) completely denervated, (3) multiply innervated, (4) thinly innervated, (5) partially innervated. A minimum of 150 NMJs were assessed blindly for each hemidiaphragm. Confocal images were obtained for publication (Leica). Differences between groups was determined using one-way ANOVA with post-hoc Bonferroni (SPSS or GraphPad Prism).

**Reporting summary**. Further information on research design is available in the Nature Research Reporting Summary linked to this article.

## Data availability
The data sets generated and/or analysed during the current study are available from the corresponding authors on reasonable request.

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

## Acknowledgements

We thank Ms. C. Mayer and Dr. K. Brock for their assistance with the animals and Ms. R. Sheth, Mr. N. Curtis, Mr. N. Staton and Ms. C. Sanapala for their support with the forelimb studies. Further, we are grateful to Dr. A. Tran, Dr. J. Cregg, Dr. M. DePaul and Dr. R. Kissane for reading the manuscript. Financial support was provided by The International Spinal Research Trust (STR117 to W.J.A., J.S. and P.M.W.), Craig H. Neilsen Foundation (221988 to W.J.A.), Wings for Life (WFL-US-027/14 to P.M.W.), the NIH (NS025713 to J.S.; R21OD018297 and R01NS101105 to W.J.A.), The Brumagin-Nelson Fund, the Hong Kong Spinal Cord Injury Fund, the Kaneko Family Fund and Unite 2 Fight Paralysis. These authors jointly supervised this work: Warren J. Alilain and Jerry Silver.

## Author contributions

All animal work and data analysis was performed by P.M.W., tissue processing and immunohistochemistry were performed by P.M.W. and S.C.S., manuscript preparation and editing was performed by P.M.W., T.E.D., P.M.M., J.S. and W.J.A. The project was conceived and designed by P.M.W., W.J.A. and J.S.

## Additional information

**Competing interests:** The authors declare no competing interests.

