## [Peer Review File · Nature Communications]

Reviewers' comments:

Reviewer #1 (Remarks to the Author):

The article by Warren et al examines respiratory function in adult rats with long term spinal cord injuries. In a series of elegant experiments, they examined the effects of a neuroplasticity-promoting enzyme (chondroitinase, ChABC) either as a single treatment, or in combination with acute repetitive intermittent hypoxia (IH) conditioning. They demonstrate that extracellular matrix breakdown and conditioning mediates restoration of respiratory function following diaphragm paralysis and, importantly, that this can occur long after spinal cord injury. This work details several ground-breaking findings: (i) that there is a continued capacity for neuroplasticity in the CNS long after spinal cord injury; (ii) that CSPGs play a vital role in masking, stabilizing and controlling activity of motor systems; (iii) that this role can be harnessed by modifying the extracellular matrix to unmask latent motor plasticity; (iv) a further conditioning intervention can synergise to maximise the potential for recovery; (v) that this approach can restore vital respiratory functions in chronic high level spinal injuries. These types of meticulous, long term experiments are very rare in the field (largely due to the costs and animal care required for long term spinal injured animals and pressure to publish precluding long term experiments), yet they are absolutely necessary. Respiratory failure is one of the leading causes of death following spinal cord injury and a major medical challenge and the implications and potential impact of this work are vast, in terms of furthering our understanding of respiratory function and physiology after trauma and exploring strategies to unmask the potential for neuroplasticity and functional repair of a vital system (which likely also expands more widely to other systems).

This paper contains an incredible amount of careful and detailed work, performed to the highest standards. The experiments are well conducted, the data solid and elegantly presented and the findings are novel and of high impact, both for the field of spinal cord repair and the wider fields of regenerative neurobiology and neuroplasticity. The authors should be commended for this impressive and top-quality body of work. I have only minor comments and suggestions, mainly to improve clarity for the readership.

1. While the injury model used is an elegant system and makes total sense for this study, providing a clean way of looking at diaphragm function and the contribution of both ipsilateral and spared contralateral systems, could the authors comment in the discussion on how this might translate to more contusive-type injuries, with bilateral damage and varying degrees of tissue sparing on either side.
2. The finding that only a subset of treated animals responded with aberrant serotonergic sprouting and tonic activity in the diaphragm is interesting and some further discussion on why this might be the case, and what may be the mechanisms, would be appreciated. The authors suggest that enzyme treatment in non-responding animals did not induce enough 5HT sprouting to facilitate positive gains in function (lines 137-139), but this seems a bit vague - was the pattern and spread of CSPG digestion quantified, and did it correlate with extent of 5HT sprouting? Was there variability in the extent of CSPG digestion in specific lamina regions of the spinal grey matter between animals, was it more lateralised or more caudal, or weaker, in the non-responders? Some more clarification would be appreciated.
3. Return of upper limb motor function (Fig S7): it is not clear exactly when these tests were performed, and from which cohort/study. Are the baseline values taken 3 months after injury and prior to ChABC administration? Were values between groups the same at baseline? It is not clear since they have all been normalised to zero. What do the data look like if actual values are shown, not

normalised data? A timeline for this behavioural data, as was nicely provided for other figures, would be appreciated and/or more clear explanation in the text. Also, the suggested mechanism for forelimb functional improvement was unmasking of perineuronal nets several segments below the site of chabc administration (at C7) – was there any evidence of digested CSPGs and/or changes in PNNs at this spinal level?

4. These are high level cervical injuries affecting breathing centres – please provide some more information on post-operative care and post-operative condition of the animals, particularly those kept for long term. What are the morbidity/mortality rates for these type of injured animals with long term breathing complications?

5. Avoid superfluous terms e.g. “uniquely” (line 27, line 405), “disappointingly” (line 50), “unprecedented” (line 62) etc. These are not necessary - the data speak for themselves.

6. Also, there is a lot of emphasis on “paralysis”, particularly in the abstract and introduction (e.g. “In spite of complete paralysis” “after a near lifetime of paralysis”). This could mislead the reader into thinking that a major focus of the study is on paralysed limbs, rather than respiratory function. Phrase differently or replace with “diaphragm paralysis”.

7. The authors might want to cite a recent nature reviews neuroscience article “breathing matters” (PMID: 29740175).

8. Line 48 talks about human SCI but cites a rat study.

9. Line 62-63 sentence ending “and both greatly superior to that which occurs acutely and constant over time” is not very clear – do you mean the effects remained constant, or that they were superior to constant treatment over time?

Reviewer #2 (Remarks to the Author):

This is an extremely interesting, well-written report of a rigorous series of experiments by leaders in the field of spinal cord injury and breathing. This is the most comprehensive assessment of respiratory function following spinal cord injury and treatment at such chronic time points. The battery of anatomical, pharmacological and electrophysiological assessments are well-described and interpreted. I greatly appreciate the attention given to variability in results, in that not all animals respond to treatments in the same way. While this work make a very important contribution to the field, there are a few minor issues that should be addressed.

Primary concerns:

The allocation of animals into experimental and control groups is confusing. Perhaps a table would be useful to address this. Based on the text, it appears that some controls were lacking. For instance, in the long-term study, only 5 animals (of 10) survived for study and they were divided into 2 separate experiment groups. Is this correct? After referring to these as Groups 1 and 2, it would be useful if the authors kept this terminology (e.g. in the next paragraph).

Recent work showed that functional outcomes can be significantly affected by anesthesia. Could the authors discuss how the functional outcomes reported in the present work could be affected by anesthesia, and the role of glutamate on recovered activity?

The authors correctly point out that spinal interneurons may play an important part in re-organization of the injured and treated phrenic network. The recent work by Cregg et al (2017) suggests that glutamatergic spinal interneurons may be an important component of this. While the authors touch on this by citing Crone et al, this is further supported by the recent results by Zhouldeva et al (2017, 2018) and Streeter et al (2018). Could the authors expand on this discussion more to include such examples?

While a remarkable degree of recovery is presented in these experiments, could the authors comment on whether a similar degree of recovery would be expected following mid-cervical SCI (particularly the contusion injury) where the spinal phrenic network is more disrupted?

Reviewer #3 (Remarks to the Author):

This study provides evidence that significant functional recovery of breathing is possible even at chronic time points following high cervical spinal cord injury. The experiments presented here follow up on previous work from the authors (Ref 10) showing that digestion of CSPGs and the PNN results in modest recovery of breathing when given acutely following SCI. They demonstrate that a single intraspinal injection of ChABC into the phrenic motor pool quickly restores diaphragm emg from completely paralyzed to near normal when given 3 months or 1.5 years post SCI. Intermittent hypoxia conditioning alone does not recover diaphragm function, but provides a modest enhancement when combined with ChABC treatment. IH and ChABC are both sufficient to increase 5HT and TrkB ipsilateral to the injury, however it seems this can only lead to functional recovery if PNN and CSPGs have been removed. The combined treatment can also transiently lead to tonic diaphragm activity that is correlated with high amounts of 5HT. Overall the results are compelling and the controls are appropriate.

Major Comments:

- 1) It has been demonstrated previously that ChABC can promote significant functional recovery following SCI. Importantly, the experiments here translate what has been shown in the locomotor systems to the respiratory control network. The most novel aspect of the current study is treatment with ChABC at chronic time-points following the injury. Although the phenomenon of restored diaphragm emg activity is robust, the experiments provide only a limited mechanistic explanation for the observed functional recovery. 5HT seems to correlate with recovery, but it does not seem to be sufficient (Fig 1D). E.g. Is 5HT necessary? Is recovery due to enhanced regeneration through the injury site or crossed phrenic pathways?
- 2) The lack of spontaneous recovery observed in this study contradicts the results of many previous studies implementing a similar C2 hemisection model. Because of these discrepancies it would be important to quantify the extent and severity of the injuries beyond what is shown in Fig S1.
- 3) Is activity of the ipsilateral diaphragm recruited by maximal drive during nasal occlusions? This should be made clear in the text and possibly displayed and quantified in a figure/supplemental figure.
- 4) Lines 109-113: The authors suggest that motor system compensation prevents the animals from being hypoxic/hypercapnic following injury. However, the contralateral diaphragm does not seem to compensate since contralateral emg activity isn't increased at any time points post injury, and at 1 week (Fig 2Cb) even seems significantly decreased (If not significant, these statistics should still be reported in the figure legend). Could this be a limitation of quantifying diaphragm activity as a percent

of maximal? Could plasticity also alter the maximal output? Please comment. Also, please make it clear how the data were normalized. Was the maximal output of the contralateral or ipsilateral emg used?

5) Lines 176-183: This section is unclear and does not appear to reflect the data shown in Fig S5. In the graphs, the differences are greatest in the Nx groups, not the IH groups and the differences are more robust for the HVR than the HCVR, contrary to what is stated in the text. As currently summarized in the text, the plethysmography data don't seem to add much substance and distract from the major conclusions of the study.

6) Throughout the figures that quantify %maximal activity of the ipsilateral diaphragm, there are many data points that exceed, and even far exceed, 100% (e.g. Figs. 2Ca, 3Ca). It is unclear how it is possible that the recorded emg activity recorded during eupnea could be higher than that achieved during maximal drive. Please explain.

7) Fig. 4C: These results should be quantified, possibly with CTA

8) The experiments are obviously time consuming and difficult. Nevertheless, the study suffers from a small number of replicates for some comparisons (e.g. Fig 4B; Fig. 6BC).

9) The manuscript contains a lot of data, and in general would benefit from revisions to the text to improve clarity and better convey the main findings of the study and their broad significance.

Minor comments:

1) E.g. stars in Fig. 1Ba: When recorded diaphragm emg was tonic, it isn't clear how the %maximal inspiratory activity could be calculated?

2) Line 103: Fractions like this are easily misinterpreted as 2 animals out of an n=3, and might be less confusing if displayed as a percentage.

3) When discussing correlations, the statistical test should be performed and p-value reported. E.g. line 99-100: for all individual animals and all conditions, did the intensity of WFA and the intensity of GFAP correlate or not when plotted as an X Y graph?

4) Lines 135-139: TrkB, WFA and 2B6 also correlate with recovery of diaphragm emg. Since serotonin can be increased by IH without inducing functional recovery (Fig 1D), aren't these other correlations more likely to be important for determining responders from non-responders? Also, please elaborate on the criterion used to distinguish responders and non-responders.

5) Line 780: E.g. Fig 2Dd, statistical results should be reported in the figure legends even if not significant.

6) Line 447: The costal diaphragm is thought to have a more dedicated respiratory function than the crural diaphragm. Why was the crural diaphragm chosen as the site for emg recording?

7) Line 499: 20 sec nasal occlusions were used to assess maximal drive. Why were only 3 breaths averaged?

8) All bar graphs should also display the individual data points.

Rebuttal to Reviewer #1:

We greatly appreciate the careful, thorough, and considered review of our manuscript, in particular that you consider the work elegant, meticulous, of a high standard, and have appreciated the novelty of both the findings drawn and rarity of such long-term experiments. Further, we wish to thank you for the time it took for you to review the manuscript and all of the comments and suggestions raised. We hope that in addressing each of these points we have facilitated in clarifying issues and statements for the reader, making a stronger and more assessable piece of work.

1) While the injury model used is an elegant system and makes total sense for this study, providing a clean way of looking at diaphragm function and the contribution of both ipsilateral and spared contralateral systems, could the authors comment in the discussion on how this might translate to more contusive-type injuries, with bilateral damage and varying degrees of tissue sparing on either side.

We wish to thank the reviewer for the opportunity to discuss the potential effects of our treatment strategy in other experimental models of respiratory dysfunction following SCI and believe that the inclusion of this information will further illustrate the potential significance of the work presented. Additions have been made to the text as requested (lines 519-527).

2) The finding that only a subset of treated animals responded with aberrant serotonergic sprouting and tonic activity in the diaphragm is interesting and some further discussion on why this might be the case, and what may be the mechanisms, would be appreciated. The authors suggest that enzyme treatment in non-responding animals did not induce enough 5HT sprouting to facilitate positive gains in function (lines 137-139), but this seems a bit vague - was the pattern and spread of CSPG digestion quantified, and did it correlate with extent of 5HT sprouting? Was there variability in the extent of CSPG digestion in specific lamina regions of the spinal grey matter between animals, was it more lateralised or more caudal, or weaker, in the non-responders? Some more clarification would be appreciated.

We are very pleased that that the reviewer finds the advent of tonic diaphragm activity or non-responders in sub-sets of our treated animals as interesting as we do. We agree that correlating the non-responders in our treatment groups with the extent of CSPG digestion is generally a good idea. However, as CSPGs and PNNs can start to return as early as 2 weeks following ChABC application, quantifying the extent of enzyme spread in our animals through IHC would be relatively inaccurate. The majority of our animals are euthanized 4 – 26 weeks post treatment. As such, the nets (particularly at the lateral and rostral/caudal extremes) may have started to return and, as we do not know the extent to which this will be happening in different regions/sections of the cord, any assessment of CSPG digestion at these time points would be inexact. However, we are of the same opinion as the reviewer that this is an important issue to consider. As such, we have further correlated our 5HT, 2B6, TrkB, WFA, and GFAP recordings to provide greater insight into the cause of the non-responders which details that this is reliant on CSPG digestion which is also correlated with the amount of 5HT at the PMP. We believe that this gives greater clarification and support to our hypothesis concerning the reason why not every animal responds to our treatment strategy. Further, we have included some additional data regarding the use of exogenously applied 5HT in generating synchronised, patterned function in the previously paralysed hemidiaphragm in ChABC treated animals (Fig. 2E; Supplementary Table 1; lines 148-159, 182-187; Supplementary material lines 132-134).

3) Return of upper limb motor function (Fig S7): it is not clear exactly when these tests were performed, and from which cohort/study. Are the baseline values taken 3 months after injury and prior to ChABC administration? Were values between groups the same at baseline? It is not clear since they have all been normalised to zero. What do the data look like if actual

values are shown, not normalised data? A timeline for this behavioural data, as was nicely provided for other figures, would be appreciated and/or more clear explanation in the text. Also, the suggested mechanism for forelimb functional improvement was unmasking of perineuronal nets several segments below the site of chabc administration (at C7) – was there any evidence of digested CSPGs and/or changes in PNNs at this spinal level?

We thank the reviewer for their comments and are very happy to provide the requested information regarding the upper limb motor function assessments. We believe that the new additional table of animal groupings (Supplementary Table S3) and timeline (Supplementary Fig. 7A) will provide the additional detail required regarding the experimental protocol and animal group used for these experiments. We are very grateful to the reviewer for pointing out the possible misinterpretation of the data regarding normalisation. In fact, the data presented are the raw figures and are not normalised. Following this injury, while the animals can walk, they do not readily use the injured ipsilateral forelimb for many activities including exploration or grooming, (which is reflected in the baseline values of these assessments). We have now made these points clearer in the text. Moreover, as per the reviewer's suggestion, we have now included within Supplementary Fig. 7 an experimental protocol detailing when the behavioural assessments were made and referred to this in the text. As per the reviewer's recommendation we have assessed the unmasking of PNNs caudal to the PNNs by showing enzyme activity which provides evidence in favour of our hypothesised mechanism for the improved forelimb function depicted within the study. We thank the reviewer for this suggestion and believe that this has helped strengthen the point being made (Supplementary Table S3; Supplementary Fig. 7A; lines 263-266, 267, 621-623; Supplementary material lines 87, 90-93).

4) *These are high level cervical injuries affecting breathing centres – please provide some more information on post-operative care and post-operative condition of the animals, particularly those kept for long term. What are the morbidity/mortality rates for these type of injured animals with long term breathing complications?*

We have expounded upon our original discussion of these points in the methods (lines 492-506).

5) *Avoid superfluous terms e.g. “uniquely” (line 27, line 405), “disappointingly” (line 50), “unprecedented” (line 62) etc. These are not necessary - the data speak for themselves.*

The text has been amended accordingly (lines 30, 53, 65, 373, and 447).

6. *Also, there is a lot of emphasis on “paralysis”, particularly in the abstract and introduction (e.g. “In spite of complete paralysis” “after a near lifetime of paralysis”). This could mislead the reader into thinking that a major focus of the study is on paralysed limbs, rather than respiratory function. Phrase differently or replace with “diaphragm paralysis”.*

The text has been amended accordingly (lines 28, 85, 343, 513). However, in the final sentence of the abstract we have not made the addition as the preceding text makes it clear we are discussing “respiratory recovery”.

7) *The authors might want to cite a recent nature reviews neuroscience article “breathing matters” (PMID: 29740175).*

The text has been amended accordingly (lines 51, and 725-726).

8) *Line 48 talks about human SCI but cites a rat study.*

The text has been amended accordingly (lines 51, and 727-728).

9) *Line 62-63 sentence ending “and both greatly superior to that which occurs acutely and constant over time” is not very clear – do you mean the effects remained constant, or that they were superior to constant treatment over time?*

The text has been amended to accurately reflect the nature of the results (lines 63-66).

Rebuttal to Reviewer #2:

We are very grateful to the reviewer for their considered review of our manuscript, in particular that you found the piece “extremely interesting”, “well-written” and determined that we have interpreted our results with accuracy and given due diligence to the variability we encountered with the data set. We very much appreciate the time the review has taken you and the issues which you raised as, we believe, in addressing these points the manuscript is a stronger piece of work.

1) The allocation of animals into experimental and control groups is confusing. Perhaps a table would be useful to address this. Based on the text, it appears that some controls were lacking. For instance, in the long-term study, only 5 animals (of 10) survived for study and they were divided into 2 separate experiment groups. Is this correct? After referring to these as Groups 1 and 2, it would be useful if the authors kept this terminology (e.g. in the next paragraph).

We apologise to the reviewer that they found our allocation of animals to experimental and treatment groups confusing and welcome the opportunity to rectify this oversight. As suggested we have included an additional table (Supplementary Table 3) which details all experimental groups, sub-groups and experimental conditions including the output measures recorded. In the ‘notes’ section of this table we have also described in detail how animal groups in our 1.5-year experiment were divided. This table has been referred to multiple times throughout the text of the manuscript in order to give greater clarification to the reader and previous group numbers have been altered (Supplementary Table 3; lines 89, 166, 200, 223, 268-269, 290-291, 307-308, 310, 341, 473-474; Supplementary material lines 149-160).

2) Recent work showed that functional outcomes can be significantly affected by anesthesia. Could the authors discuss how the functional outcomes reported in the present work could be affected by anesthesia, and the role of glutamate on recovered activity?

There is a large body of data starting from the late 1800’s detailing the effect of anaesthesia upon respiratory muscle and pulmonary function. We are excited to be given this opportunity to discuss this work within our methods section and have made additions to the text accordingly (lines 581-588).

Please see our response to point 3) which addresses your concerns regarding the discussion of glutamate within the text.

3) The authors correctly point out that spinal interneurons may play an important part in re-organization of the injured and treated phrenic network. The recent work by Cregg et al (2017) suggests that glutamatergic spinal interneurons may be an important component of this. While the authors touch on this by citing Crone et al, this is further supported by the recent results by Zhouldeva et al (2017, 2018) and Streeter et al (2018). Could the authors expand on this discussion more to include such examples?

We thank the reviewer for their comments. Due to the limitations of space and reference number we did not feel able to elaborate on this topic within the original document. We have now made the suggested alterations to the text, albeit briefly, to further comment on this important mechanistic component of respiratory motor function (lines 406-413 and 795-803).

4) While a remarkable degree of recovery is presented in these experiments, could the authors comment on whether a similar degree of recovery would be expected following mid-cervical SCI (particularly the contusion injury) where the spinal phrenic network is more disrupted?

We wish to thank the reviewer for the opportunity to discuss the potential effects of our treatment strategy in other experimental models of respiratory dysfunction following SCI and

believe that the inclusion of this information will further illustrate the potential significance of the work presented. Additions have been made to the text as requested (lines 519-527).

Rebuttal to Reviewer #3:

We wish to thank the reviewer for their attentive assessment of our manuscript, in particular that you find the results “compelling”. We greatly appreciate the time that such comprehensive assessment would have taken. We believe that in addressing these comments the work has become both much clearer and convincing.

Major Comments:

1) It has been demonstrated previously that ChABC can promote significant functional recovery following SCI. Importantly, the experiments here translate what has been shown in the locomotor systems to the respiratory control network. The most novel aspect of the current study is treatment with ChABC at chronic time-points following the injury. Although the phenomenon of restored diaphragm emg activity is robust, the experiments provide only a limited mechanistic explanation for the observed functional recovery. 5HT seems to correlate with recovery, but it does not seem to be sufficient (Fig 1D). E.g. Is 5HT necessary? Is recovery due to enhanced regeneration through the injury site or crossed phrenic pathways?

We thank the reviewer for their comments and the opportunity to discuss this matter at greater length. We have included additional evidence of the necessity of 5HT to directly cause activity of the ipsilateral hemidiaphragm within Fig. 2E which further develops the data already presented through the IHC analysis. We have also advanced this argument illustrating that increases in 5HT correlate with both the activity of the ipsilateral hemidiaphragm and removal of CSPGs and the PNN about the PMP (and subsequently ChABC activity). Furthermore, we have established the mechanism of the tonic activity within the paper (specifically the receptor sub-types involved), and we believe that this acts in combination with our other data to show the necessity of 5HT in producing ipsilateral diaphragm recovery after injury. Of course, we are aware the other molecules may be important in the recovery process. To this effect, we have developed our discussion to include a more complete examination of the affect glutamate might have within the system.

The persistent recovery induced was not caused by regeneration of pathways through the injury site, as can be seen through the addition of injury site volumetrics, conducted weeks following treatment application (Supplementary Fig. 2C+D). These data show that that injury remains complete regardless of time or treatment. This was expected, as we did not apply treatments at the level of the injury (aiding regeneration) but at the PMP (to mediate recovery of function). We have added text to the manuscript to make this point clearer. Further, we have added passages to the manuscript describing that the recovery engendered may have been caused through the turning on of the crossed phrenic pathways, or could be through extensive sprouting from these pathways on the contralateral side of the cord. This fully describes the anatomical circuitry responsible for the complete and persistent recovery of function demonstrated in our extended chronic model. Any further examination of this system would require the use of genetic tools in mouse models, and thus the validation and elucidation of the precise anatomical circuitry of neurons in a different model system using DREADDs or optogenetic strategies. This is something we are currently working on but we do not believe that it is necessary for inclusion within this manuscript given the robust functional data and mechanistic data provided (Fig. 2E; Supplementary Fig. 1C+D, 2C+D; Supplementary Table 1; lines 78-79, 148-159, 182-187, 397-399, 412-413, 665-671; Supplementary material lines 23-25, 132-134)

2) The lack of spontaneous recovery observed in this study contradicts the results of many

previous studies implementing a similar C2 hemisection model. Because of these discrepancies it would be important to quantify the extent and severity of the injuries beyond what is shown in Fig S1.

The figures and text have been amended accordingly (Supplementary Fig. 1B-D; lines 72-78, 665-671; Supplementary material lines 23-25).

3) Is activity of the ipsilateral diaphragm recruited by maximal drive during nasal occlusions? This should be made clear in the text and possibly displayed and quantified in a figure/supplemental figure.

The figures and text have been amended accordingly (Supplementary Fig. 1E; lines 79-84; Supplementary material lines 25-27).

4) Lines 109-113: The authors suggest that motor system compensation prevents the animals from being hypoxic/hypercapnic following injury. However, the contralateral diaphragm does not seem to compensate since contralateral emg activity isn't increased at any time points post injury, and at 1 week (Fig 2Cb) even seems significantly decreased (If not significant, these statistics should still be reported in the figure legend). Could this be a limitation of quantifying diaphragm activity as a percent of maximal? Could plasticity also alter the maximal output? Please comment. Also, please make it clear how the data were normalized. Was the maximal output of the contralateral or ipsilateral emg used?

We thank the reviewer for bringing this to our attention. When we referred to compensatory activity in the text we would not have expected this to be shown in EMG activity of the contralateral hemidiaphragm. Instead we were referring to the cited reference (Goshgarian et al., 1986) who showed that compensatory activity occurs after injury through a change in respiration rate. We have amended the text of the manuscript to add clarity to this point. The reviewer is correct that the amplitude of the contralateral hemidiaphragm minimally decreases 1 week following injury, however, this is not statistically significant compared to baseline. The text has been amended accordingly to make this point clear (see minor comments point 5). Nonetheless, the reviewer is correct at recognizing this tendency and we propose it may perhaps be an artefact of recovery from the recent surgery rather than a limitation of the commonly applied method of EMG analysis (e.g. Polentes et al., 2004, Mantilla et al., 2011, Felix et al., 2014, Seven et al., 2014). This method of analysis was selected as it makes our data comparable with that of the literature (e.g. Mantilla et al., 2011, Felix et al., 2014, Seven et al., 2014, Gill et al., 2015) and is a way of ensuring accuracy of data reporting is maintained despite minor alterations in electrode placement between animals, and removes potential differences in relative amplitude based on the depth of anaesthesia between animals (although the same dose for weight is given to each animal). We recognise that there are limitations with every way that one may choose to normalise data. However, we believe that this is the most accurate method for our experimental design and is grounded within the literature. We have further clarified the text regarding the normalisation of the data set to the hemidiaphragm muscles (lines 122, 591).

5) Lines 176-183: This section is unclear and does not appear to reflect the data shown in Fig S5. In the graphs, the differences are greatest in the Nx groups, not the IH groups and the differences are more robust for the HVR than the HCVR, contrary to what is stated in the text. As currently summarized in the text, the plethysmography data don't seem to add much substance and distract from the major conclusions of the study.

We apologise that our explanation of the data lacked clarity. We have substantially modified the text to better reflect the data obtained and accurately convey the conclusions drawn. However, we disagree with the reviewer that the plethysmography data does not add much substance. The strength of this data set is that it shows that improvements are present in freely behaving animals, without the limitations associated with anaesthesia. Nonetheless, lack of differences between these groups does not minimize the results obtained from the

EMG recordings, as their strength allows us to isolate the beneficial effects of treatment to a specific neural pathway. In including both sets of data we are showing that the treatment strategy has persistent effects at all levels within the respiratory motor system and we have amended the text to address this point. We believe that this data as a whole is valuable to the complete understanding of recovery for the reader (lines 204-216).

6) Throughout the figures that quantify %maximal activity of the ipsilateral diaphragm, there are many data points that exceed, and even far exceed, 100% (e.g. Figs. 2Ca, 3Ca). It is unclear how it is possible that the recorded emg activity recorded during eupnea could be higher than that achieved during maximal drive. Please explain.

We thank the reviewer for this comment and the opportunity to discuss it further. This is a potential caveat of using normalisation for analysis and we see that this has only happened on three data points, one each in figs. 2Ca, 3Ca and 3Cb. If the hemidiaphragm is working at its maximum capacity (which the data accurately describes) then (regardless of the extent to which drive is increased) amplitude does not really increase with nasal occlusion. As we are averaging the activity in both the occlusion and during eupnoea, statistically the average value during normal breathing can result in being slightly higher than that shown in the occlusion and thus greater than 100% of the value. This has been shown in other papers using a similar method (e.g. Felix et al., 2014; Seven et al., 2018 – this can be determined through looking at the reported SEM, p value and n number). We recognise that it is a caveat associated with this analysis method (which other groups have published with previously). However, analysis using this method is common within the field (e.g. Mantilla et al., 2011, Felix et al., 2014, Seven et al., 2014, Gill et al., 2015) and it allows us to compare our data with that which has been published, ensures accuracy of data reporting is maintained despite minor alterations in electrode placement between animals, and eradicates potential differences in relative amplitude based on the depth of anaesthesia between animals (although the same dose for weight is given to each animal). Through showing all our individual data points we are trying to be very open and clear about the variance (and any caveats) within our data set. Further, our data still accurately reflects what occurred within our animals – these results demonstrate that the animals were breathing at their maximal capacity during eupnoea, and slightly higher than that during challenge. This method of analysis is the most accurate and clear way of reporting our data and we have tried to represent the whole data set clearly within our figures.

7) Fig. 4C: These results should be quantified, possibly with CTA.

The text and figure have been amended accordingly (Fig. 4C; lines 332-333, 920-922).

8) The experiments are obviously time consuming and difficult. Nevertheless, the study suffers from a small number of replicates for some comparisons (e.g. Fig 4B; Fig. 6BC).

We appreciate the reviewer's appreciation of the length and difficulty of the experiments we have presented in this manuscript. Long term experiments of this nature are very rare in the field, but we believe they are necessary to understand the optimal ways to treat spinal injury and (ultimately) facilitate the translation of experimental and basic science to clinical application. In rebuttal to the point raised, we would like to direct the reviewer to one of the opening passages within our methodology. "*Power analysis was conducted prior to all experiments to ensure n numbers were sufficient to yield reliable data*". The power analysis conducted was within the 88 to 96 percentiles. This included our 1.5 year cohort of animals (fig. 6BC) where time and resources were prohibitive of the use of larger animal numbers. In the one instance where power was not sufficient, statistical analysis was not conducted and we only discuss the trends in our data set (this occurred for fig. 4B). However, we never considered statistically analysing this selected grouping as it is incorrect to perform statistical

analysis on a group when they have been subdivided based upon an outcome measure. We have reported (and have statistical power) for the intensity readings of immunohistochemistry for the group as a whole. We discuss the trends in the subdivided groups (and do so in that context, without implication that we have statistically analysed them). These trends were subsequently validated by pharmacological experiments, which also did have sufficient power. We have further altered the methodology to reflect these points in greater depth (lines 446-469).

9) The manuscript contains a lot of data, and in general would benefit from revisions to the text to improve clarity and better convey the main findings of the study and their broad significance.

We are grateful that the reviewer appreciates the amount of work and data that has gone into this manuscript. Since the reviewer last read the piece (and as a result of the review process) we have made substantial additions and modifications to the manuscript which we believe will aid clarification for the reviewer and we hope this satisfies their comment. However, following discussion with the authors, we agree that the findings and implications of this work have been reported clearly and accurately. This is a view point our other reviewers seem to be in agreement with describing the piece as an elegant piece of work and well written. As such, without the further explanation as to the exact part of the text that the reviewer wishes expounded upon or altered, we anticipate the changes we have now made will satisfy the reviewers request.

Minor comments:

1) E.g. stars in Fig. 1Ba: When recorded diaphragm emg was tonic, it isn't clear how the %maximal inspiratory activity could be calculated?

As is shown through CTA (Figs. 4A+C, and 5), and described within the text of our manuscript, the tonic activity in our ipsilateral hemidiaphragm has structure, particularly demarcations for the initiation of inspiration and expiration. As such, the length and the maximum and minimum amplitude of each breath during eupnoea and nasal occlusion could easily be determined. Further, tonic activity increases in amplitude during nasal occlusion similar to 'normal' breathing. Consequently, the percentage maximal tonic inspiratory activity could be calculated without difficulty. The manuscript has been amended to add clarity to this issue (lines 594-596).

2) Line 103: Fractions like this are easily misinterpreted as 2 animals out of an n=3, and might be less confusing if displayed as a percentage.

The text has been amended accordingly (lines 111, 135, 172, 288-289, 344, 350).

3) When discussing correlations, the statistical test should be performed and p-value reported. E.g. line 99-100: for all individual animals and all conditions, did the intensity of WFA and the intensity of GFAP correlate or not when plotted as an X Y graph?

With the greatest of respect, we should like to direct the reviewer to Supplementary Table 1 where all this information has been detailed. Nonetheless, out of deference to the reviewer's preference, we have now elaborated upon this information in the manuscript text (lines 146-147, and 148-156).

4) Lines 135-139: TrkB, WFA and 2B6 also correlate with recovery of diaphragm emg. Since serotonin can be increased by IH without inducing functional recovery (Fig 1D), aren't these other correlations more likely to be important for determining responders from non-responders? Also, please elaborate on the criterion used to distinguish responders and non-

responders.

We are grateful to the reviewer for discussing this issue. While TrkB is likely to be important to recovery, the amount of TrkB on the ipsilateral PMP does not correlate with the degree of (and thus not related to recovery or plasticity induced by the ChABC treatment) it is less likely than 5HT to facilitate recovery alone. We have elaborated upon this point within the text to ensure that this issue is clearly explained. Further, we have amended the text in the methods to describe how we define responders from non-responders (lines 148-159, and 598-599).

5) Line 780: E.g. Fig 2Dd, statistical results should be reported in the figure legends even if not significant.

Due to the limitations of space within the figure legend, the results of all ANOVAs/t-tests have been reported in the manuscript text. The results of the post-hoc tests have been graphically shown in the figures and reported in the associated legend. We have amended the text of the figure legends to reflect that if no comparison is graphically shown in the figure, the result was not significant (lines 893, 902, 911, 938-939; Supplementary materials lines 35-36, 43-44, 58-59, 71, 78-79, 95, 115, and 127)

6) Line 447: The costal diaphragm is thought to have a more dedicated respiratory function than the crural diaphragm. Why was the crural diaphragm chosen as the site for emg recording?

The crural diaphragm has been used in numerous papers from multiple different groups to record diaphragm EMG recordings (for example: Trelease et al., 1982; Nantwi et al., 1998; Buttry and Goshgarian, 2014; Alilain et al., 2011; Awad et al., 2013) and as such we were following established, published protocols in assessing motor activity through this region. Further, we disagree that, in the context of our experimental model, there will be any difference between the respiratory output of the crural and costal regions of the diaphragm. The mechanical properties and fibre type composition of both regions are statistically identical (Metzger et al., 1985). Both regions are similarly affected by mechano-stimulation (Jammes et al., 2000) and are involved in respiratory activity. The muscle itself is not organised that different sections are innervated by specific spinal roots of the phrenic nerve (Hammond et al., 1989) meaning that there is no anatomical arrangement of the diaphragm which corresponds to spinal arrangement. Of course, the crural region of the diaphragm stops respiratory activity briefly during swallowing and oesophageal distension (Cherniack et al., 1984; Altschuler et al., 1985), however, these activities would not occur while the animal is under anaesthesia, which is when our EMGs were recorded. As such, the crural region of the diaphragm represents an optimal position from which to record functionality. This being the case, we were not complacent to the multiple regions of the diaphragm while assessing functional output. In all animals, if no EMG signal was achieved upon first placement of the electrodes we tried in a number of places in the crural region to obtain a signal. We then checked in the sternal and costal regions of the muscle for function. Animals which showed no EMG output in the crural region showed the equivalent response in all other areas of the hemidiaphragm. Electrodes were then tested to ensure they were functioning correctly and guarantee accurate data collection. We have now altered the text in our methods section to reflect this (lines 569-572).

7) Line 499: 20 sec nasal occlusions were used to assess maximal drive. Why were only 3 breaths averaged?

Assessing diaphragm EMG amplitude normalised to nasal occlusion is typical in the field to overcome issues associated with electrode placement (e.g. Polentes et al., 2004, Mantilla et al., 2011, Felix et al., 2014, Seven et al., 2014). However, there is divergence within the field as to how this should be calculated. In our animals, performing an occlusion for longer than

20 seconds was not an ethical or practical assessment as it typically resulted in an apnoea that could lead to the death of the animal. Many groups assess breaths at the end of the nasal occlusion. However, we noted in many animals that the breaths at the end of the occlusion were not always the maximal evoked (fig. S1E contralateral). Further, with many animals we noted that breath length and cycle time would substantially increase during the occlusion (especially towards its end point). As such, giving a time frame in which to assess breaths (e.g. 5 seconds from the end) would only result in the assessment of 1 or 2 sub-maximal breaths. The rolling average we employed is an accurate system used within the field (e.g. Baker-Herman et al, 2010) to assess the average maximal response over a given time frame and optimised maximal output measures. Of course, it takes a substantial number of breaths for an animal to reach maximal output during occlusion (Supplementary Fig. 1E). Further, our animals make far fewer breaths during this period. As such, after substantial quantification and assessment of the nasal occlusion from a series of preliminary animals, we determined that the average of three breaths gave the most accurate assessment of maximal diaphragm output over the 20 sec nasal occlusion. Reference to this has now been made in the text (Supplementary Fig. 1E; Supplementary materials lines 25-27).

8) *All bar graphs should also display the individual data points.*

The authors chose not to present data in this fashion in the original manuscript as it makes many of the figures seem much more complex and thus can reduce clarity. However, we have amended the figures at the reviewer's request (Figs. 1Dav, 1Dbv, 1Dcv, 1Ddv, 2Dd, 3Dd, 4Ba+b, 6Dd; Supplementary Figs. 2A, 2Bav, 2Bbv, 2Bcv, 2Bdv, 2Bev, 2Bfv, 3Ah-i, 3B, 4Ba-e, 4Ca-e, 5Aa-c, 5Ba-c, 6Ad, 9Ad+e, 9B-D, 10Ba+b, 10Caiv, 10Cbiv, 10Cciv, 10Cdiv, 10Ceiv, 10Cfiv, 10Cgiv).

REVIEWERS' COMMENTS:

Reviewer #1 (Remarks to the Author):

I have no further concerns and I commend the authors for their careful to addressing all of the reviewer comments. This is an excellent paper.

Reviewer #2 (Remarks to the Author):

The authors have invested a great deal of effort in editing the manuscript for resubmission, and it is clearly reflected in this revision. I commend the them on thoroughly addressing the issues raised and strengthening the manuscript overall.

Two very minor points:

1. Zhouldeva et al (2018) actually used rats, not mice.
2. There had been some confusion in the mechanism of plasticity being described. Perhaps the authors could clarify their discussion of compensatory plasticity by distinguishing between neural (neurogram or EMG) and behavioural (ventilation) compensatory plasticity, as defined by Kleim (2012) and Hoh et al (2013).

Reviewer #3 (Remarks to the Author):

The authors have done an excellent job revising this manuscript. The additional data included, and discussion of the various points brought up by the reviewers, has improved the overall clarity and impact of the study. My general and specific comments and concerns have been sufficiently addressed in the text and rebuttal. This reviewer has no further concerns and recommends this study for acceptance.

Reviewer #1:

I have no further concerns and I commend the authors for their careful to addressing all of the reviewer comments. This is an excellent paper.

We are exceptionally grateful to the reviewer for their comments and suggestions on the piece and pleased that they consider the amendments and rebuttals we have made to all the reviewers comments worthwhile. We thank you for this appraisal of the work and the paper.

Reviewer #2:

The authors have invested a great deal of effort in editing the manuscript for resubmission, and it is clearly reflected in this revision. I commend the them on thoroughly addressing the issues raised and strengthening the manuscript overall.

Two very minor points:

- 1. Zhauldeva et al (2018) actually used rats, not mice.*
- 2. There had been some confusion in the mechanism of plasticity being described. Perhaps the authors could clarify their discussion of compensatory plasticity by distinguishing between neural (neurogram or EMG) and behavioural (ventilation) compensatory plasticity, as defined by Kleim (2012) and Hoh et al (2013).*

We thank the reviewer for all their comments and for the continued evaluation of the work, which we believe adds greater clarity to the text and are pleased that you believe the previous amendments made have strengthened the manuscript. In response to your most recent points:

1. We apologise that in the editing of your manuscript the reference to Zhauldeva *et al* (2018) being performed in rats was inadvertently removed. This has now been amended.
2. In the three instances we discuss compensatory activity with the results of our manuscript we do so in the context of developing the discussion of our EMG recordings, as such the type of plasticity being referred to is clear e.g. "compensatory increases in contralateral diaEMG amplitude". In the one instance where the type of compensatory plasticity may not have been evident we have added clarity by discussing the "behavioural compensation".

Reviewer #3:

The authors have done an excellent job revising this manuscript. The additional data included, and discussion of the various points brought up by the reviewers, has improved the overall clarity and impact of the study. My general and specific comments and concerns have been sufficiently addressed in the text and rebuttal. This reviewer has no further concerns and recommends this study for acceptance.

We are thrilled that you consider the alterations we have made to the text will improve clarity and impact of the piece and that you are satisfied with our responses to your specific points. We appreciate your recommendation for acceptance of the work.